# Short-term interaction between silent and devastating earthquakes in Mexico

V. M. Cruz-Atienza [1✉], J. Tago[2], C. Villafuerte [3], M. Wei[4], R. Garza-Girón[5], L. A. Dominguez [6], V. Kostoglodov[1], T. Nishimura [7], S. I. Franco [1], J. Real[1], M. A. Santoyo[1], Y. Ito[7] & E. Kazachkina[3]

Either the triggering of large earthquakes on a fault hosting aseismic slip or the triggering of slow slip events (SSE) by passing seismic waves involve seismological questions with important hazard implications. Just a few observations plausibly suggest that such interactions actually happen in nature. In this study we show that three recent devastating earthquakes in Mexico are likely related to SSEs, describing a cascade of events interacting with each other on a regional scale via quasi-static and/or dynamic perturbations across the states of Guerrero and Oaxaca. Such interaction seems to be conditioned by the transient memory of Earth materials subject to the "traumatic" stress produced by seismic waves of the great 2017 (Mw8.2) Tehuantepec earthquake, which strongly disturbed the SSE cycles over a 650 km long segment of the subduction plate interface. Our results imply that seismic hazard in large populated areas is a short-term evolving function of seismotectonic processes that are often observable.

[1] Instituto de Geofísica, Universidad Nacional Autónoma de México, Mexico City, Mexico. [2] Facultad de Ingeniería, Universidad Nacional Autónoma de México, Mexico City, Mexico. [3] Posgrado en Ciencias de la Tierra, Universidad Nacional Autónoma de México, Mexico City, Mexico. [4] Graduate School of Oceanography, University of Rhode Island, Narragansett, USA. [5] Department of Earth and Planetary Sciences, University of California, Santa Cruz, USA. [6] Escuela Nacional de Estudios Superiores, Universidad Nacional Autónoma de México, Morelia, Mexico. [7] Disaster Prevention Research Institute, Kyoto University, Kyoto, Japan. ✉email: cruz@geofisica.unam.mx

The seismicity rate varies over time and depends on changes in both the state of stress and properties of the solid Earth. The diversity of earthquakes discovered in recent years, together with new observations of very small transient variations in the crustal properties, offer an unprecedented perspective for exploring causality between different seismotectonic processes. Inferred effects of slow slip events (SSE, also called silent earthquakes) on large and devastating earthquakes have led to critical questions closely related to seismic hazard. The role of SSEs in the seismic cycle seems to have been preponderant in the initiation of some megathrust earthquakes[1–5]. Observations also show that transient waves from teleseismic or regional earthquakes may trigger SSEs and tectonic tremor[6–10], which are two closely related phenomena in active faults. Highly pressurized fluids where slow earthquakes happen[11] make frictional conditions very sensitive to small stress or strain perturbations[12,13], thus playing an important role in the generation of SSEs and, certainly, in their interaction with devastating events.

Recently, three major earthquakes took place in southcentral Mexico causing more than 480 deaths and losses of 1,6 billion dollars. The earthquake sequence initiated with the great Mw8.2 Tehuantepec event on September 8, 2017, the largest earthquake ever recorded in Mexico, which may have broken the whole subducted Cocos lithosphere[14,15] (Fig. 1). Eleven days later and 480 km northwest, on September 19, the Mw7.1 Puebla-Morelos normal-faulting (57 km depth) event delivered a deadly shock to Mexico City[16], where 44 buildings collapsed and 600 were seriously damaged despite its remarkably slow, dissipative rupture[17]. The sequence ended five months later on February 16, 2018, with an Mw7.2 thrust event below Pinotepa Nacional, Oaxaca (hereafter Pinotepa), more than 250 km away from both previous earthquakes, causing damage where similar ruptures have severely harmed local infrastructures in the past. Besides damaging earthquakes, the Mexican subduction zone is prone to very large SSEs and persistent tectonic tremor, especially in the Guerrero and Oaxaca states, which extend along the epicentral regions of the earthquake sequence[18–27]. At the time of the Tehuantepec and Puebla-Morelos events, two separate SSEs were taking place in Guerrero and Oaxaca[26–28]. As we will discuss later

on, other SSEs also happened in both states in an unusual way during and after the five-month earthquake sequence, featuring a unique story that deserves to be told and understood.

In this work, we investigate possible interactions between such SSEs and the three devastating earthquakes and found that most of our observations can be explained as a regional cascade of causally related events through short-term, quasi-static and dynamic interactions that have strongly perturbed the regional SSE cycles in the states of Guerrero and Oaxaca.

## Results

**Aseismic slip history of the plate interface.** In the Mexican subduction zone, slow surface displacement can be explained in terms of the aseismic slip between the subducted Cocos plate and the overriding North American plate. Such slip can be understood either as SSEs, post-seismic relaxations, or plate interface coupling (PIC, i.e., $1 - v/b$, where $v$ is the interplate slip rate, $b$ is the plate convergence rate, and $v \leq b$). For imaging the spatial evolution of the aseismic slip in those terms, we inverted continuous displacement records at 57 permanent GPS stations from November 2016 to October 2019, the largest dataset ever analyzed in Mexico, making use of ELADIN, a recently developed and powerful technique[29] (see "Methods" section, Supplementary Fig. 1). Careful examination of the GPS time series revealed several transient deformations in the Guerrero and Oaxaca states. Figure 2 presents the aseismic-slip inversion results for the whole analyzed period, where we find: (Fig. 2A) an almost typical interseismic deformation period; (Fig. 2B) the 2017 Mw6.9 Guerrero SSE (G-SSE1) that reached shallow interface regions (up to 10 km depth, Supplementary Fig. 2) and the initiation of the 2017 Oaxaca SSE (O-SSE1) before the onset of the earthquake sequence; (Fig. 2B–D) the evolution of the Mw6.9 O-SSE1; (Fig. 2E, F) the Mw7.2 post-seismic slip of the Pinotepa earthquake (PE-afterslip) that lasted at least until November 2018, together with a neighboring but separated, 200 km length, Mw6.9 SSE in Guerrero (G-SSE2, second one); and (Fig. 2G, H) the concomitant evolution of the 2019 Mw7.0 Guerrero (G-SSE3, the third one) and Mw6.9 Oaxaca (O-SSE2, second one) SSEs

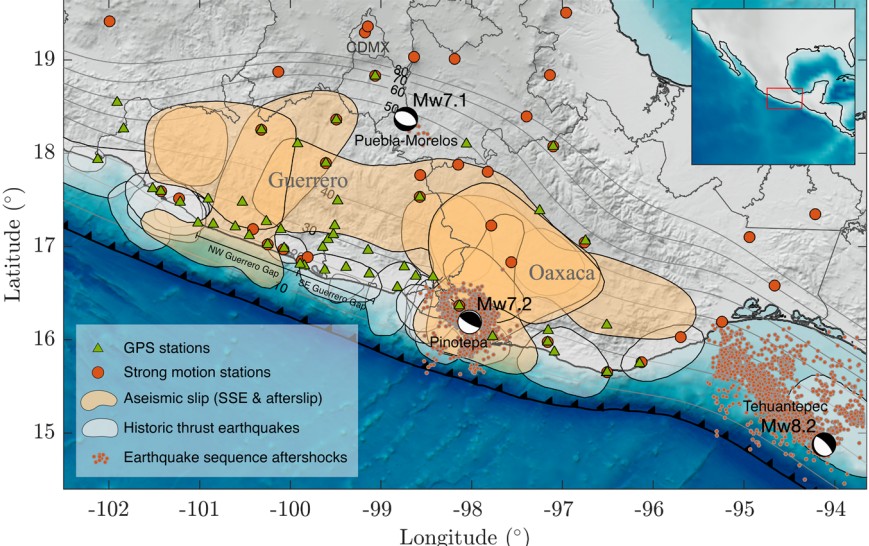

**Fig. 1 Study region and regional instrumentation around the Tehuantepec (Mw8.2), Puebla-Morelos (Mw7.1), and Pinotepa (Mw7.2) earthquake sequence.** Orange shaded areas depict the 1 cm aseismic slip contours imaged between June 2017 and July 2019 in the plate interface. Green triangles and orange circles indicate GPS and strong motion sites, respectively. White shaded areas delineate rupture zones of historic thrust earthquakes. Orange dots show the 10-days aftershock sequences as reported by the SSN except for the Mw7.1 earthquake, for which three-months aftershocks are reported. Gray contours show iso-depths (in kilometers) of the 3D plate interface and CDMX denotes Mexico City.

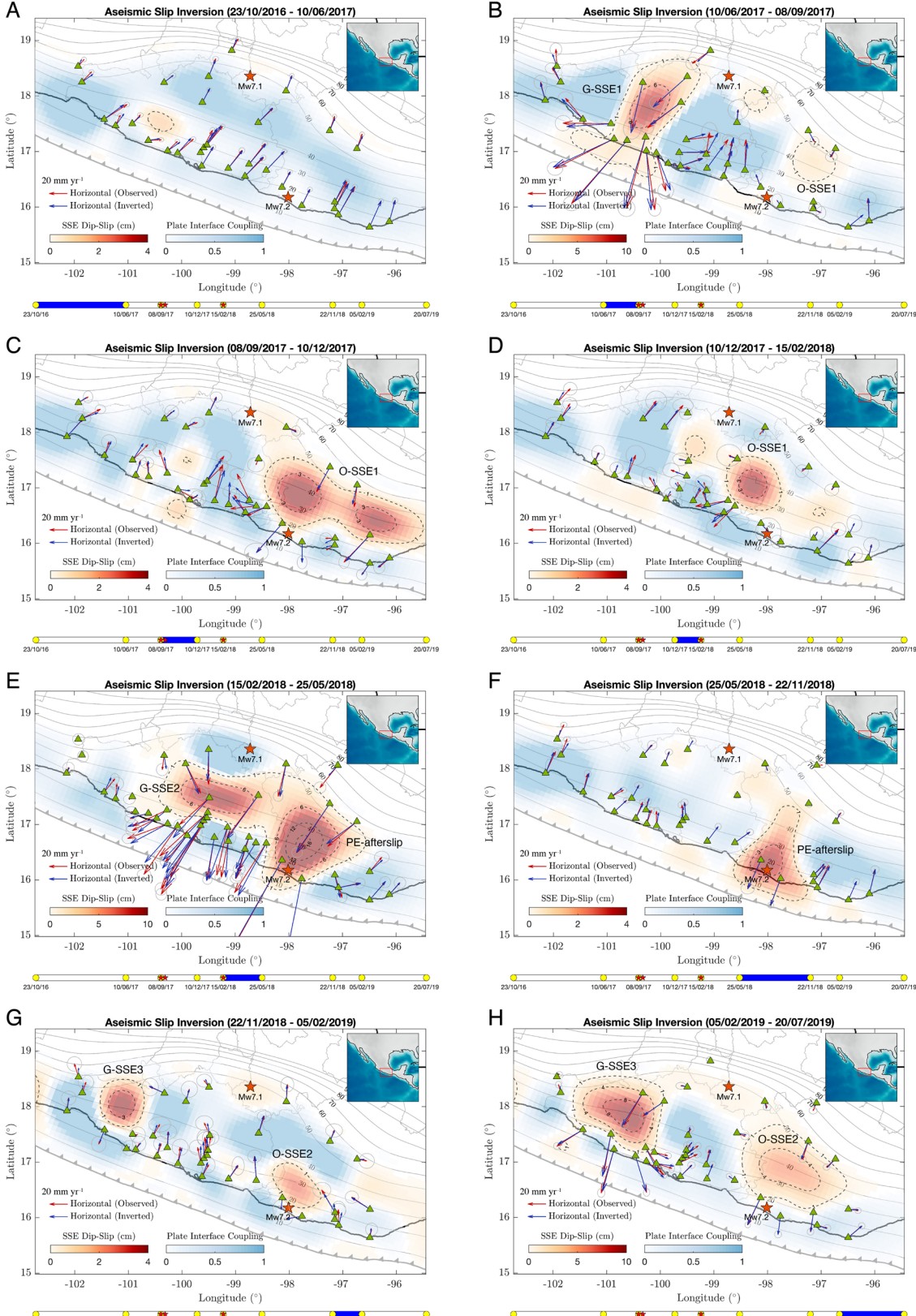

(Table 1). The aseismic slip evolution for all time windows is summarized in Fig. 3A and integrated into Supplementary Movie 1, where we display the whole space-time evolution of the events interpolated linearly every 30 days. Considering only the slip areas encompassed by 1 cm contours (Figs. 1 and 3A), the aseismic moment released during this three-year period is equivalent to a magnitude Mw7.5 earthquake ($M_0 = 2.32 \times 10^{20}$ Nw m), where only 31% of $M_0$ corresponds to the afterslip of the Mw7.2 Pinotepa rupture (Table 1).

Figure 4 shows the aseismic slip evolution (for events with Mw > 6) throughout the period of the earthquake sequence. For the analysis, we separated the slip history into two parts; one before

**Fig. 2 Aseismic slip inversions for the whole analyzed period across and after the earthquake sequence (see also Fig. 3 and Supplementary Movie 1).** We find (**A**) an almost typical inter-seismic deformation period; (**B**) the 2017 Guerrero SSE (G-SSE1) and the initiation of the 2017 Oaxaca SSE (O-SSE1); (**B–D**) the evolution of the O-SSE1; (**E–F**) the post-seismic slip of the Mw7.2 Pinotepa earthquake (PE-afterslip) together with a neighboring but separated SSE in Guerrero (G-SSE2, second one); and (**G–H**) the concomitant evolution of the 2019 Guerrero (G-SSE3, third one) and Oaxaca (O-SSE2, second one) SSEs (see Table 1). Dashed slip contours are in centimeters. Yellow circles encompassing the blue bar at the bottom of each panel indicate the dates of the associated inverted window, and red small stars, the Mw8.2 Tehuantepec, Mw7.1 Puebla-Morelos, and Mw7.2 Pinotepa earthquakes timing, respectively, from left to right. Red and blue arrows show the observed and synthetic surface horizontal displacements, and the gray ellipses one standard deviation of the corresponding GPS data window.

**Table 1 Dates (dd/mm/yy) and moment magnitudes (Mw) estimated from the 1 cm slip contours of all aseismic slip events reported in this work.**

| Guerrero | | | Oaxaca | | |
|---|---|---|---|---|---|
| **Event** | **Dates** | **Mw** | **Event** | **Dates** | **Mw** |
| G-SSE1 | 10/06/17–08/10/17 | 6.91 | O-SSE1 | 01/06/17–15/02/18 | 6.93 |
| G-SSE2 | 16/02/18–01/06/18 | 6.93 | PE-afterslip | 16/02/18–22/11/18 | 7.17 |
| G-SSE3 | 22/11/18–20/07/19 | 6.99 | O-SSE2 | 05/02/19–20/07/19 | 6.92 |

The prefixes G and O refer to the states of Guerrero and Oaxaca, respectively, while PE refers to the Pinotepa earthquake.

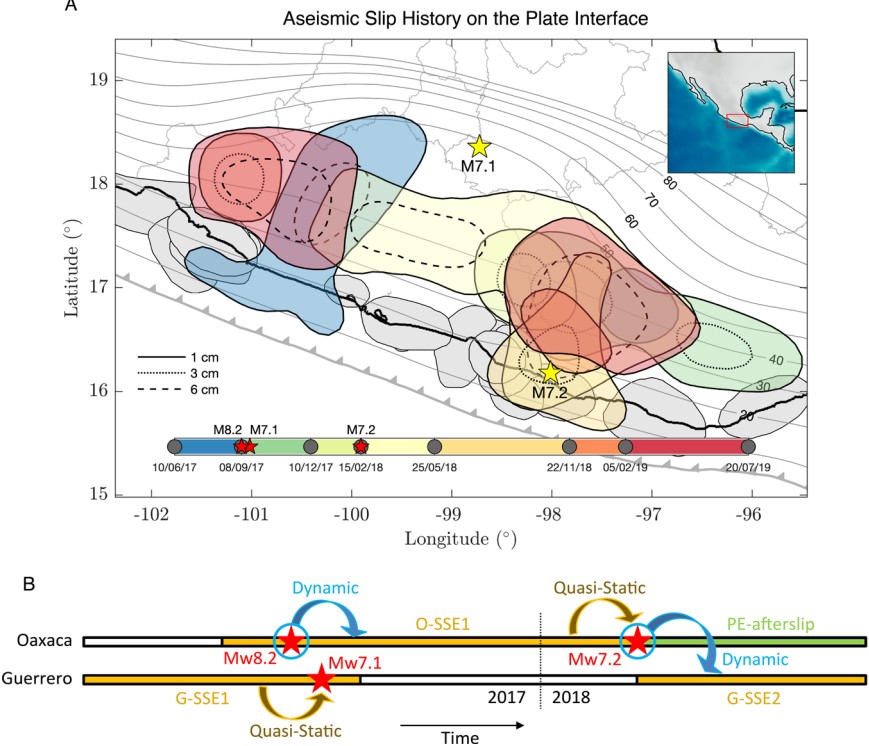

**Fig. 3 Evolution of the aseismic slip at the plate interface and types of interaction between the different events. A** Aseismic slip patches are those of Figs. 1 and 2 but color-coded according to the timespan of each event (see colorbar). Gray contours show iso-depths (in kilometers) of the 3D plate interface and yellow stars the epicenters of the Puebla-Morelos (Mw7.1) and Pinotepa (Mw7.2) earthquakes. **B** Sketch showing the evolution of events across the earthquake sequence in the states of Guerrero and Oaxaca, and the nature of the interaction between them, either dynamic or quasi-static.

(Fig. 4A) and the other after (Fig. 4B) the Pinotepa earthquake. The second part includes the previous inverted window as a reference. Panel A (and the GPS time series at the ARIG station in panel B, left) shows that the G-SSE1 basically ended with the occurrence of the devastating Mw8.2 Tehuantepec and Mw7.1 Puebla-Morelos earthquakes. Only a few minor slip patches were imaged in the following three months (Fig. 2C). We further see that the O-SSE1, which also initiated months before the earthquakes, developed bilaterally during the five months that

followed the sequence initiation. More interestingly, the examination of the GPS time series in the southern stations reveals a sudden reversal of the displacement direction from north to south (green circles, left) when the great Tehuantepec event took place. In contrast, northern stations (green circles, right) feature a slow, typical SSE initiation well before, around May–June 2017. The sharp change of the deformation regime in the south suggests that the Tehuantepec earthquake modified the ongoing Oaxaca SSE. The question also arises as to whether the Guerrero and Oaxaca

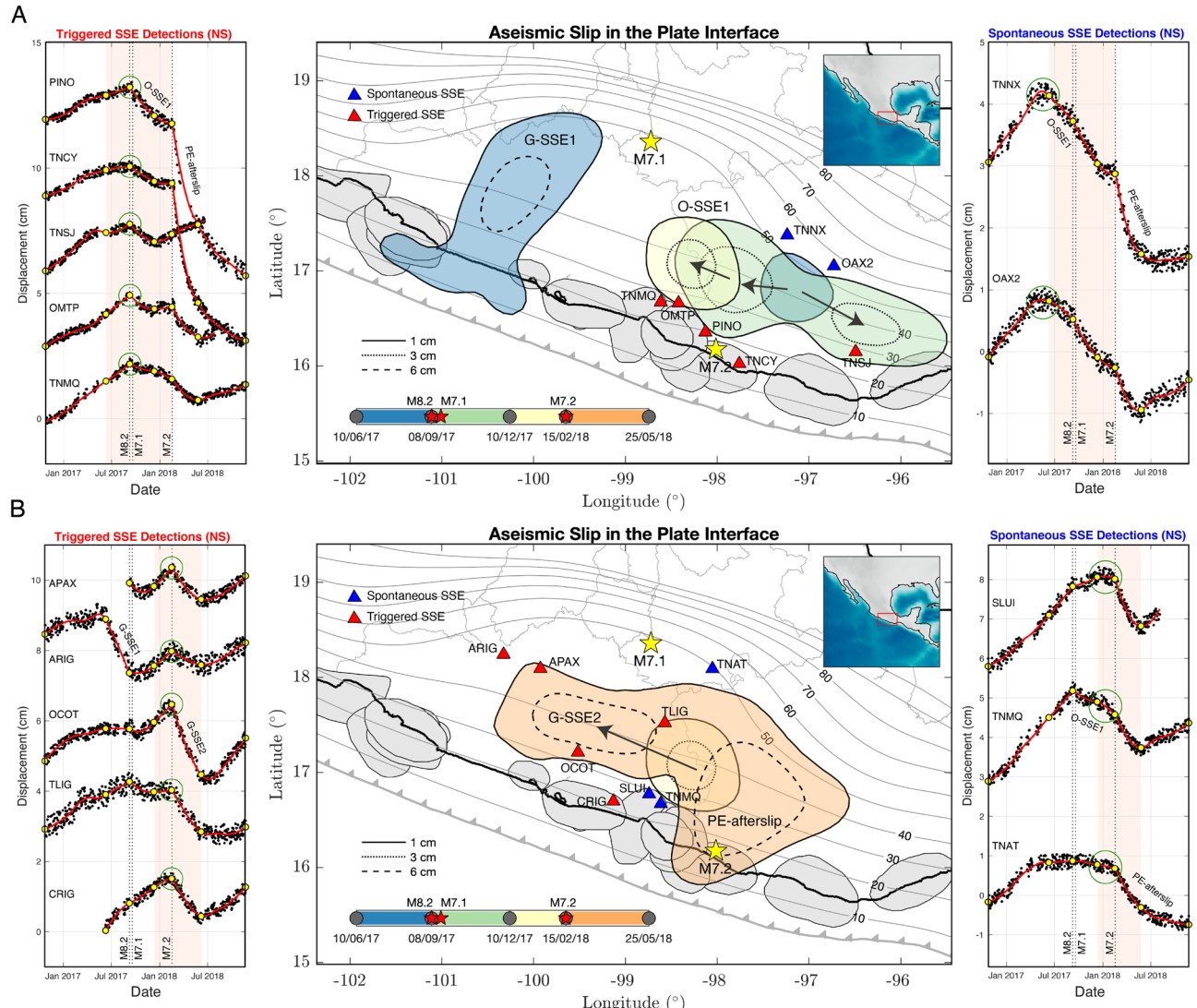

**Fig. 4 Evolution of the plate interfaces aseismic slip (SSEs and afterslip) during the earthquake sequence (separated in two parts) and representative GPS time series (north-south components).** The first part (panel **A**) before the M7.2 Pinotepa earthquake and the second part (panel **B**) after the earthquake. Pink shaded rectangles encompass the GPS windows (yellow dots) that were slip-inverted in the central maps (color areas) for each panel. Blue triangles show GPS stations where we observe spontaneously initiated or preexistent SSEs at the time of earthquakes (right panels, green circles), while red triangles show the stations where we observe triggered SSEs also at those times (left panels, green circles). Notice the abrupt reversal of the deformation pattern in the left panels (from north to south, green circles) right at the moment of the Mw8.2 Tehuantepec and Mw7.2 Pinotepa earthquakes. Gray contours show iso-depths (in kilometers) of the 3D plate interface and gray shaded regions the rupture areas of historical earthquakes.

SSEs could have promoted the rupture of the Puebla-Morelos and Pinotepa events, respectively, as proposed for other earthquakes in Mexico[5,23].

The GPS displacements in panel B show a similar effect over the ongoing Oaxaca SSE to that inferred for the Tehuantepec earthquake, but in this case, produced by the Mw7.2 Pinotepa event. While displacements in the eastern stations show either an ongoing or a smooth, spontaneously initiated SSE before this earthquake (green circles, right), some stations to the west exhibit again an abrupt change of displacements from north to south, right when the earthquake happened (green circles, left).

All reported SSEs (i.e., three in Guerrero and two in Oaxaca) and the PE-afterslip overlap one another outlining a 650 km long, trench-parallel band of aseismic stress release (Figs. 1 and 3A). Effects of the earthquakes on the SSE activity or, inversely, of the SSEs on the earthquakes' initiation may have occurred due to static and/or dynamic stress/strain perturbations. In the following, we examine these possibilities.

**Stress transfer and seismicity**. Stress transfer to active faults has long been recognized as a preponderant factor in earthquake occurrence[30]. Although fault failure depends on the absolute stress level, changes of the Coulomb Failure Stress (CFS) can explain rupture sequences and seismicity-rate variations remarkably well. CFS changes smaller than 50 kPa are often spatially well correlated (above 65%) with triggered seismicity and significantly larger (one order of magnitude) than values required for triggering slow earthquakes in subduction zones[31].

The 1 cm slip contour of the G-SSE1 stopped about 80 km from the Puebla-Morelos intraslab earthquake hypocenter (Fig. 2B). The CFS on the seismogenic fault (i.e., within a 20 km radius from the hypocenter) due to the plate-interface aseismic slip evolution (SSE + PIC) reveals a rise of 35 kPa around the earthquake hypocenter in the 40 days preceding the rupture (Supplementary Figs. 3A–D and 3E, see "Methods" section). Albeit this increment is in the upper part of the 10–50 kPa earthquake triggering range commonly referred to in

the literature[30] and similar to the one believed to have triggered the Mw7.3 (2014) Papanoa earthquake by an SSE in Guerrero[5], interestingly, it occurred in the late stage of the SSE, when the PIC near the rupture area experienced a recovery certainly affected by the neighboring SSE evolution. This unexpected behavior of the interface coupling during an SSE has also been observed in the last three SSEs in Oaxaca[32], the last one only two months before the recent Huatulco earthquake (Mw7.4) of June 23, 2020, suggesting that an interaction exists between different interface regions experiencing either stress-release or stress build-up. As we will discuss later, the strong shaking produced in the seismogenic fault by the great Tehuantepec earthquake eleven days earlier, could significantly reduce the intraslab fault strength[33,34] and thus anticipate the Mw7.1 Puebla–Morelos rupture initiation[35] driven by the CFS induced by the aseismic slip at the plate interface (i.e., by the SSE and the associated PIC changes) (Fig. 3B). To our knowledge, this is the first evidence that an SSE-related process in the plate interface could promote the initiation of a devastating intraslab rupture such as the Puebla-Morelos earthquake.

Five months later, the Mw7.2 Pinotepa thrust earthquake took place at the Cocos–North American plate boundary (Fig. 1) while the O-SSE1 was unfolding (Fig. 4A). The detailed aseismic slip and CFS evolution on the plate interface preceding the earthquake are shown in Supplementary Fig. 4. Around the hypocentral region, there is a clear rise of CFS reaching cumulative values close to 400 kPa (Fig. 5A). During the five months following the Mw8.2 Tehuantepec rupture and within a radius of 20 km from the Pinotepa earthquake hypocenter, the CFS experienced a sustained growth of 200 kPa due to the SSE development to the north (Fig. 5B). During the same period, GPS inversions show that the interplate slip rate, which always remained in a coupling regime (i.e., smaller than the plate convergence rate), decreased until the initiation of the earthquake (i.e., the PIC increased from 0.1–0.2 to ~0.65). However, the area north of the hypocenter, where maximum seismic moment was released during the Pinotepa earthquake (between 20 and 30 km depth)[36], was indeed pervaded by the O-SSE1 with a slip of 1 to 3 cm (Fig. 4A). To better elucidate the mechanical process leading to the Pinotepa earthquake nucleation, we carefully analyzed the seismicity in the hypocentral region during the year preceding the event using two complementary template matching techniques (see "Methods" section, Supplementary Figs. 5 and 6). Figure 5C shows 21-day event counts with a magnitude larger than 2.1 and foci within a 30 km radius from the hypocenter. Notice the outstanding spatial correlation between the CFS concentration and the precursor seismicity next to the earthquake hypocenter (inset of Fig. 5A). Our seismic catalog has 431% more detections (5977 earthquakes) than those reported by the Servicio Sismológico Nacional (SSN) above the completeness magnitudes for the same period and hypocentral distance. One clear characteristic stands out from the temporal evolution of our earthquake catalog: seismicity raised steadily after the Mw8.2 Tehuantepec event until the Mw7.2 Pinotepa earthquake, especially during the two previous months (up to ~50% increase) when the O-SSE1 induced the largest CFS increment in the hypocentral region (see also Supplementary Fig. 4F).

The increase in CFS, PIC and seismicity rate in the hypocentral region before the Pinotepa earthquake strongly suggests that the dominant mechanism that led to the onset of rupture corresponds to an asperity model; i.e., a heterogeneous initial stress in the source region was loaded at a mesoscale by the development of the SSE to the north until an overloaded nucleation patch, the asperity (e.g., subducted seamount), overcame the plate interface strength. Despite the increasing coupling of the plate interface (and CFS) during the preparedness of the earthquake, seismicity also increased next to the hypocenter. This

scenario disfavors the putative widespread idea of an SSE-induced aseismic slip acceleration around the nucleation patch, observed for other large earthquakes[1,2], as the main triggering mechanism for this event (Fig. 3B). The small magnitude precursor seismicity reveals small-scale processes that cannot be resolved by our GPS inversions. However, this activity can be explained by a cascading rupture of small, neighboring asperities loaded by the mesoscale effect of the SSE evolution north of the hypocenter.

In addition, except for the large PE-afterslip area and the very eastern portion of the O-SSE1, static CFS perturbations produced by the earthquake sequence seem not to have had a major bearing on the SSE activity as can be appreciated in Fig. 6B, D, where positive stress values in most areas of the subsequent SSEs (green contours) are negligible.

**Plate interface dynamic perturbations**. Abrupt changes in the slow crustal deformation pattern after the Tehuantepec and Pinotepa earthquakes (Fig. 4) suggest an effect of both events on the interplate aseismic slip that cannot be explained by static stress transfers, as shown in the last section. However, dynamic stress or strain perturbations produced by seismic waves may have important implications in the elastic properties of fault zone materials (e.g., transient reduction of the bulk modulus) and the slip behavior, especially where slow earthquakes take place[6,7,9,33,35,37]. For instance, long-period surface waves from the 2010 Mw8.8 Maule earthquake-triggered deep tremor in Guerrero and likely reactivated an ongoing SSE[8].

We estimated dynamic perturbations at the plate interface for both earthquakes of the sequence (see "Methods" section). Figure 6A shows the CFS peak values produced by the Rayleigh waves (25 s period) of the Mw8.2 Tehuantepec event (Supplementary Figs. 7 and 8) beneath strong-motion stations in south-central Mexico. Dynamic perturbations around the O-SSE1 region lasted about 80 s and are characterized by three major wave cycles with CFS values ranging between 75 and 200 kPa, and absolute dilations between 1.4 and 6.0 microstrain (Supplementary Fig. 9). Albeit the dynamic triggering of slow earthquakes also depends on the (uncertain) preexistent fault condition, dynamic dilations from the Tehuantepec event are two orders of magnitude larger than those produced in Japan by the great Sumatra–Andaman 2004 earthquake, which triggered widespread tremor in Shikoku and Tokai regions[6] and CFSs about eight times larger[38]. The earthquake triggered tremor in Oaxaca[26] and Jalisco[39], and an SSE in the San Andreas fault[10], 3000 km northwest from the source. Since the O-SSE1 initiated before the earthquake and considering that tremor sensitivity increases as the slow slip develops[40], it is plausible that such dynamic perturbations were responsible for the large SSE enhancement and thus of the sudden change of the crustal deformation pattern in the region (Figs. 4A and 3B).

Given that the Mw7.2 Pinotepa earthquake is a much smaller event that occurred closer to the (presumably) triggered G-SSE2 (Fig. 2B), shorter-period body waves could also affect the SSE that was unfolding in Oaxaca at the moment of rupture. Figure 6C shows the complete-wavefield CFS maximum values simulated on the plate interface for the earthquake using the DGCrack numerical platform[41] (see "Methods" section, Supplementary Fig. 10). Values range between 100 and 150 kPa within the G-SSE2 slip area, where prestress increments were already above 400 kPa due to the O-SSE1 (Fig. 5A), and overcome 400 kPa in the post-seismic slip region downdip from the epicenter. In contrast, the co-seismic static CFS change produced by the earthquake is at least two orders of magnitude smaller in the same SSE region (Fig. 6D). This indicates that seismic waves of the Pinotepa earthquake could also be responsible for triggering the second SSE in Guerrero (G-SSE2) and

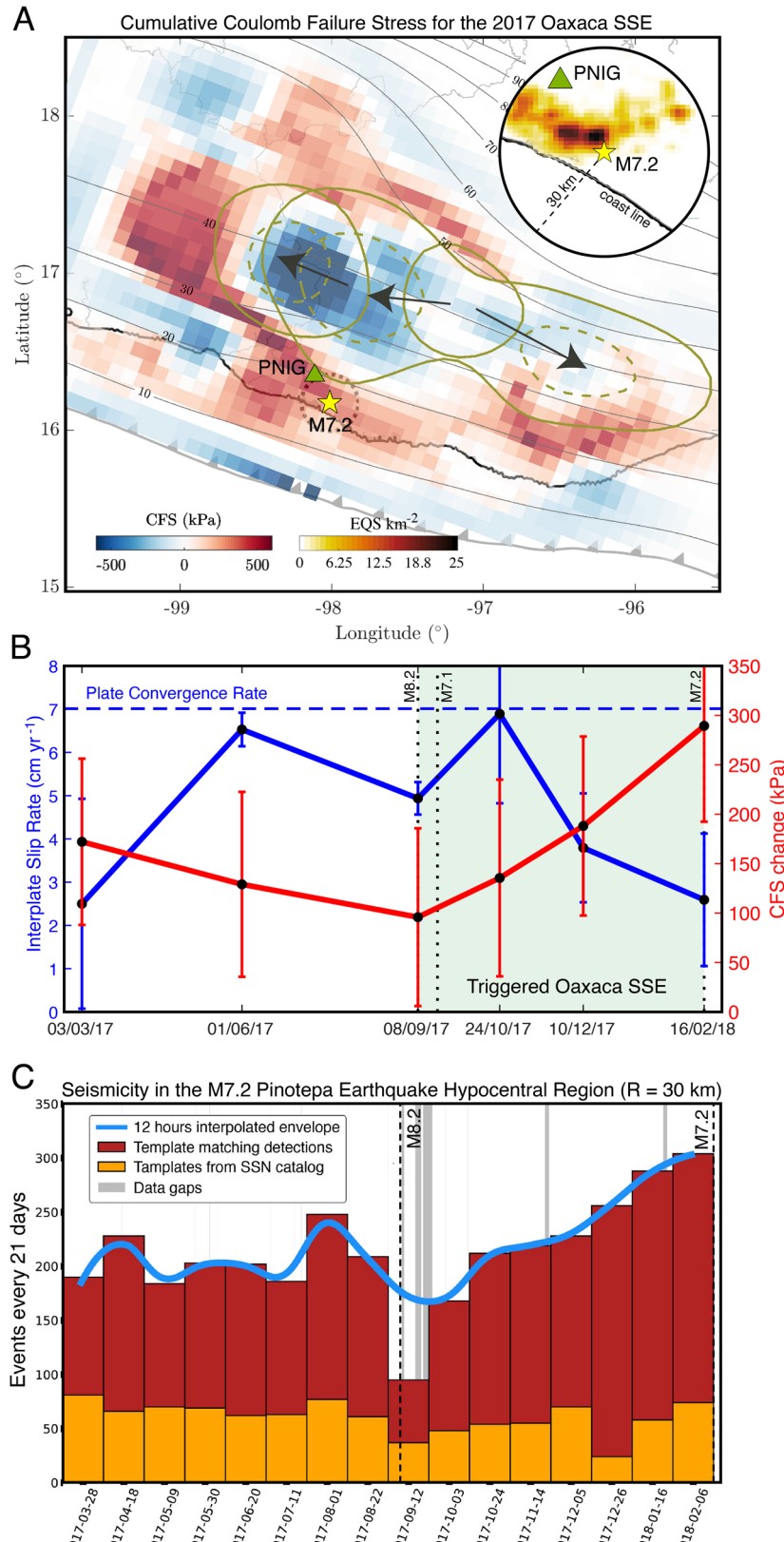

therefore the change in the regional deformation pattern at the time of the event (Figs. 4B and 3B).

**Mechanics of SSEs dynamic triggering**. To assess whether seismic waves from the Tehuantepec and Pinotepa earthquakes could explain the abrupt changes of the crustal deformation pattern, we conducted numerical simulations of SSEs in the framework of rate-and-state (R&S) friction models subject to the stress dynamic perturbations estimated for both earthquakes. Previous studies with similar methods[10,42] focused on dynamically triggered SSEs when the perturbation occurs in the inter-SSE period. However,

**Fig. 5 Coulomb Failure Stress (CFS), Plate Interface Coupling (PIC), and seismicity rate evolution before the Pinotepa earthquakes in the vicinity of its hypocenter. A** 15-month cumulative CFS on the plate interface and spatial evolution of the O-SSE1 (1 cm slip solid contours and 3 cm slip dashed contours). The density of the template matching earthquake detections is shown in the inset (i.e., of the precursor seismicity). Gray contours show iso-depths (in kilometers) of the 3D plate interface and the green triangles the broadband seismic station PNIG. **B** Temporal evolution of the CFS change and the interplate slip rate averaged within a 20 km radius from the Pinotepa earthquake hypocenter (dotted circle, panel **A**) along with the associated standard deviations (vertical bars). See also Supplementary Fig. 4. **C** Seismicity rate evolution for template matched events (M > 2.1) within a distance $R = 30$ km from the Pinotepa earthquake hypocenter (see Supplementary Figs. 5 and 6).

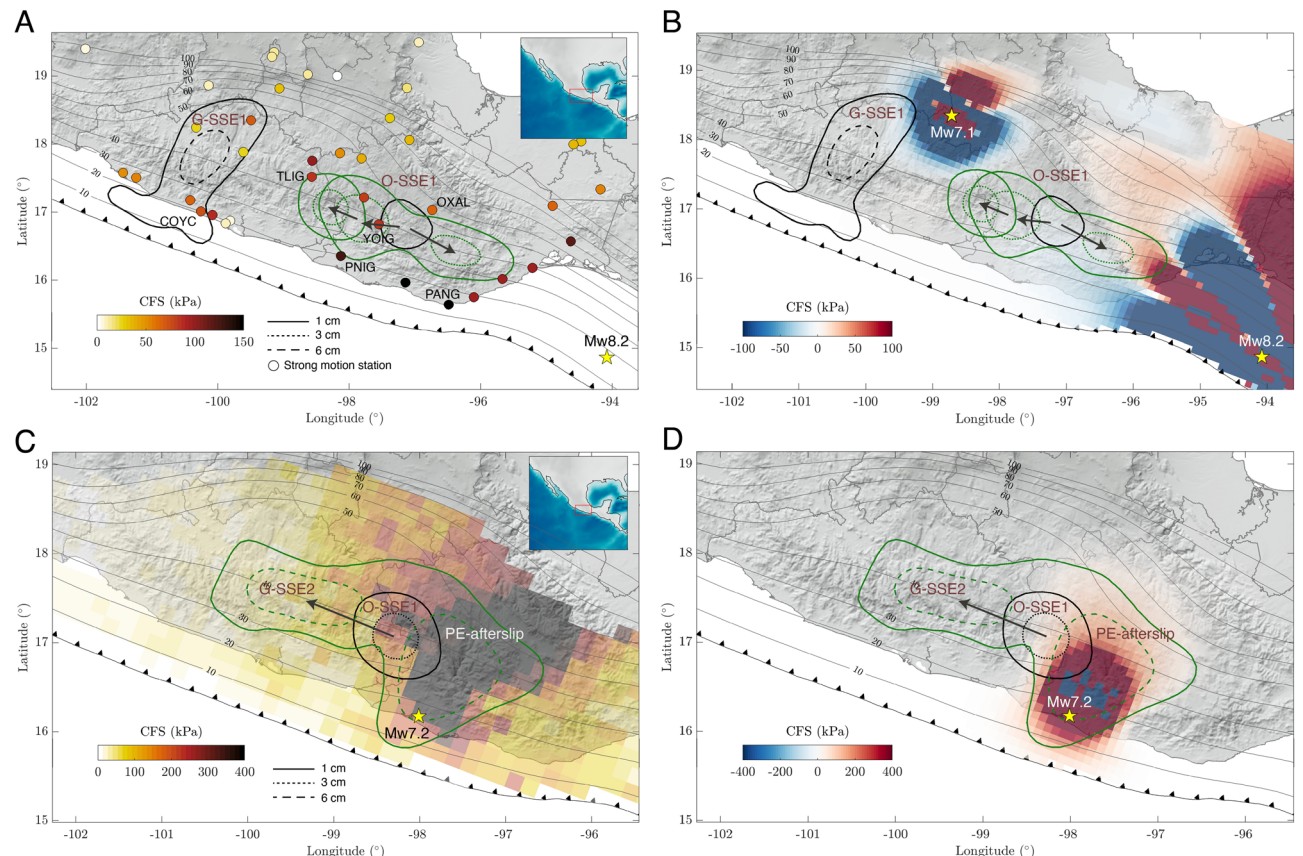

**Fig. 6 Dynamic (peak values) and static Coulomb Failure Stresses (CFS) on the 3D plate interface (gray contours in kilometers) produced by the Mw8.2 Tehuantepec (A and B, respectively) and Mw7.2 Pinotepa (C and D, respectively) earthquakes in the plate convergence direction for a friction coefficient of 0.5.** Aseismic slip events right before the corresponding earthquake are shown with black contours, while those that occurred immediately after the earthquake are shown with green contours. Dynamic stresses for the Tehuantepec event (**A**) were computed from actual strong motion records at different sites (colored circles, see Supplementary Fig. 9A). Estimates for the Pinotepa event (**C**) were computed and validated by means of a 3D Discontinuous Galerkin finite-source numerical simulation of the earthquake using the DGCrack platform[41] (see Supplementary Fig. 10).

the Tehuantepec and Pinotepa earthquakes happened during the large O-SSE1 (Fig. 4), making this a unique opportunity to better understand the mechanics of SSEs when seismic waves from M7 + and larger regional earthquakes perturb them in a tectonic environment where both phenomena are frequent.

Following Wei et al.[42], we developed a 2D R&S SSE model for the Oaxaca region (Fig. 7A) (see "Methods" section, Supplementary Fig. 11). Figure 7C shows the model response to dynamic stresses estimated for the Tehuantepec earthquake at the plate interface under station YOIG, which is located above the O-SSE1 slip area (Fig. 6A and Supplementary Fig. 9). The final slip due to the stress perturbation is about twice the value of the reference, spontaneous SSE. Figure 7B shows the "aseismic slip jump" induced by this perturbation, where the propagation speed of the SSE front experiences an abrupt acceleration which, in turn, implies a change of the same order in the surface displacements. The higher the CFS peak value of the perturbation, the larger are both the final slip and the SSE front and slip accelerations. The

same happens with the perturbations estimated for the Pinotepa earthquake (Fig. 7D). However, despite that peak values over the O-SSE1 region are significantly larger than those induced by Rayleigh waves from the Tehuantepec event (>250 kPa, Fig. 6C), they overcome the SSE triggering threshold for a much shorter time (intense phase durations for the Mw8.2 and Mw7.2 events are ~75 s and ~13 s, respectively). Consequently, the slip increment associated with each wavelet exceeding the threshold is smaller. This is clear in the insets of Fig. 7C, D, where the slip rate response and cumulative slip increment due to several waves from the Pinotepa earthquake are comparable to the increment of a single phase of the Tehuantepec event. Therefore, the dominant period of seismic waves also controls its SSE triggering potential and thus the effective fault response (Supplementary Fig. 11D). Since our model considers only along-dip SSEs propagation and the actual slip in Oaxaca and Guerrero migrated predominantly along-strike, it is clear that seismic waves from both earthquakes could produce a much longer SSE evolution than theoretically

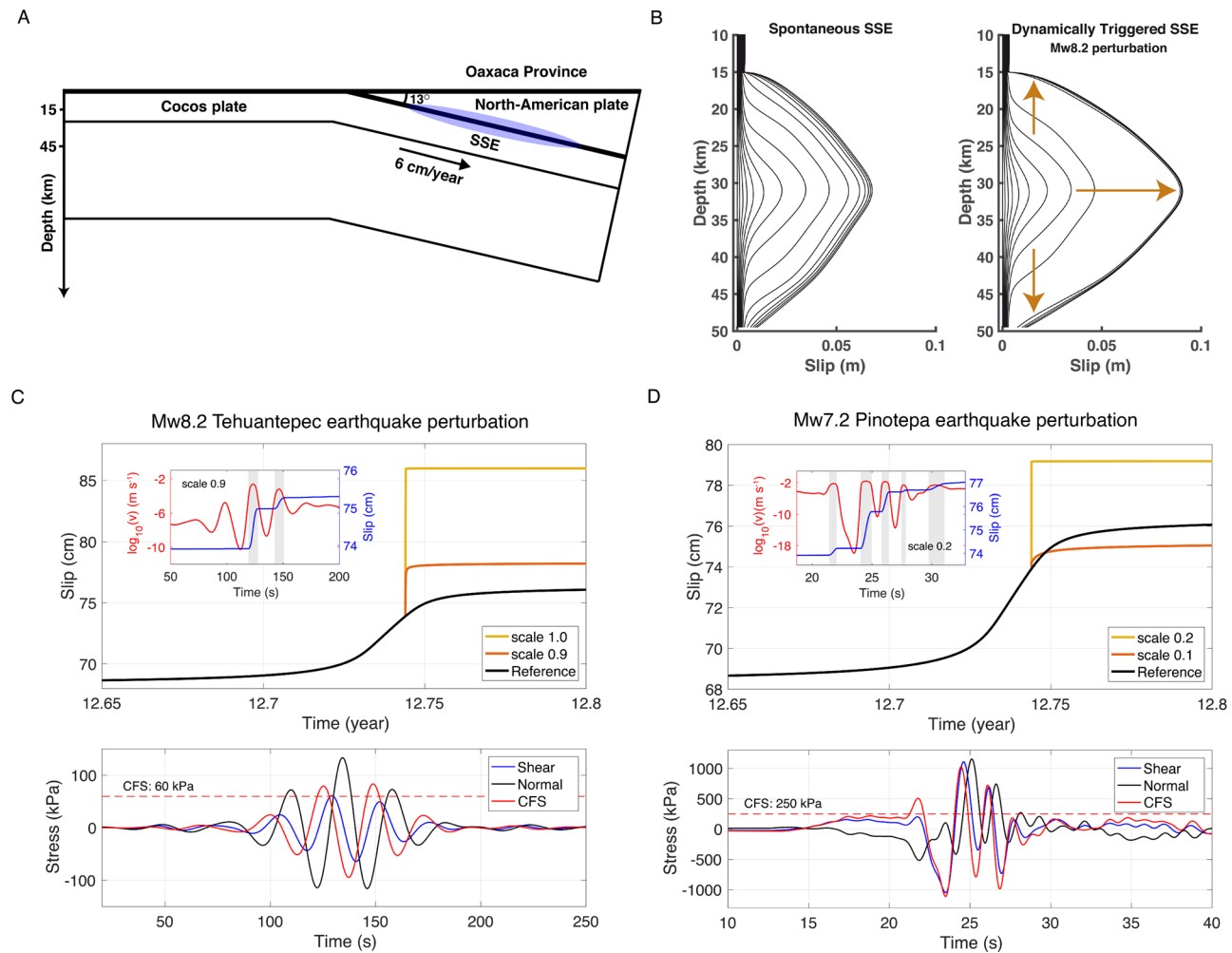

**Fig. 7 Rate-and-state fault models for SSE triggering by seismic-wave stress perturbations. A** Synoptic 2D model of the subduction zone in the study area. **B** Slip evolution of a spontaneous SSE and a dynamically triggered SSE in the R&S friction model subject to the Mw8.2 Tehuantepec earthquake stress perturbations estimated under the YOIG station (Fig. 6A and Supplementary Fig. 9). The contours time increment is about 2 days. **C** Top, slip evolution of the SSE reference model, and two triggered events at 31 km depth for stress perturbations due to the Mw8.2 Tehuantepec earthquake with different scaling factors. The inset shows the slip velocity and slip at that depth for the 0.9 scaled perturbation. Bottom, unscaled stress perturbation used in these simulations. **D** the Same as **C** but for the Mw7.2 Pinotepa earthquake. Please note that the scaling factors are different.

predicted by our simple model, explaining thus the observed crustal rebounds initiated with both ruptures (Fig. 4).

## Discussion

During two years, between June 2017 and July 2019, in addition to the devastating earthquake sequence, five large SSEs (Mw > 6.9) occurred in southcentral Mexico over a trench-parallel continuous band of 650 km in length with a cumulative moment magnitude Mw7.4 (Figs. 1 and 3A, Table 1). Three of them in Guerrero, and the other two in Oaxaca interspersed by the Pinotepa earthquake post-seismic slip with Mw7.2 (Fig. 3B). Among all aseismic events, only the 2017 Guerrero and Oaxaca SSEs (G-SSE1 and O-SSE1) initiated before the earthquake sequence, so that 87% of the total aseismic moment was released during the 1.7 years following the great Mw8.2 Tehuantepec rupture, when the earthquake sequence started. Although the three Guerrero SSEs nucleated in different regions (Fig. 2), all of them overlap downdip of the Northwest Guerrero seismic gap with a slip larger than 5 cm each (Figs. 1, 3 and Supplementary Fig. 2). Unlike the last 20 years, during which all SSEs occurred every ~4 years in Guerrero (six events between 1998 and 2017)[5],

the last two events reported here had much smaller recurrence periods, of 0.25 and 0.5 years for the G-SSE2 and G-SSE3, respectively. Figure 8 shows a detailed comparison of displacement time series at different GPS sites in Guerrero, including the longest record in Mexico, from CAYA station, since 1997. A simple inspection of that record reveals the clear disruption of the SSE cycle in that province after the Mw8.2 Tehuantepec event.

Something unusual also happened in Oaxaca; the plate interface slipped (aseismically) continuously for the whole two years period with at least two reactivations, one during the post-seismic relaxation of the Mw7.2 Pinotepa earthquake, and the other one around December 2018, when the O-SSE2 initiated. Figure 8 further shows the long record at PINO station, where we appreciate how the return period of SSEs in Oaxaca was also reduced after the Mw8.2 earthquake. This is clear when comparing the 8 months between O-SSE2 and O-SSE3, the later event (not studied here) starting two months before the Mw7.4 Huatulco earthquake of June 23, 2020[32], and the ~1.5 years that typically elapse between the silent events in Oaxaca[23]. It is worth mentioning that despite the data scarcity at PINO station between 2007 and 2012, the SSEs indicated in the figure (vertical blue bars)

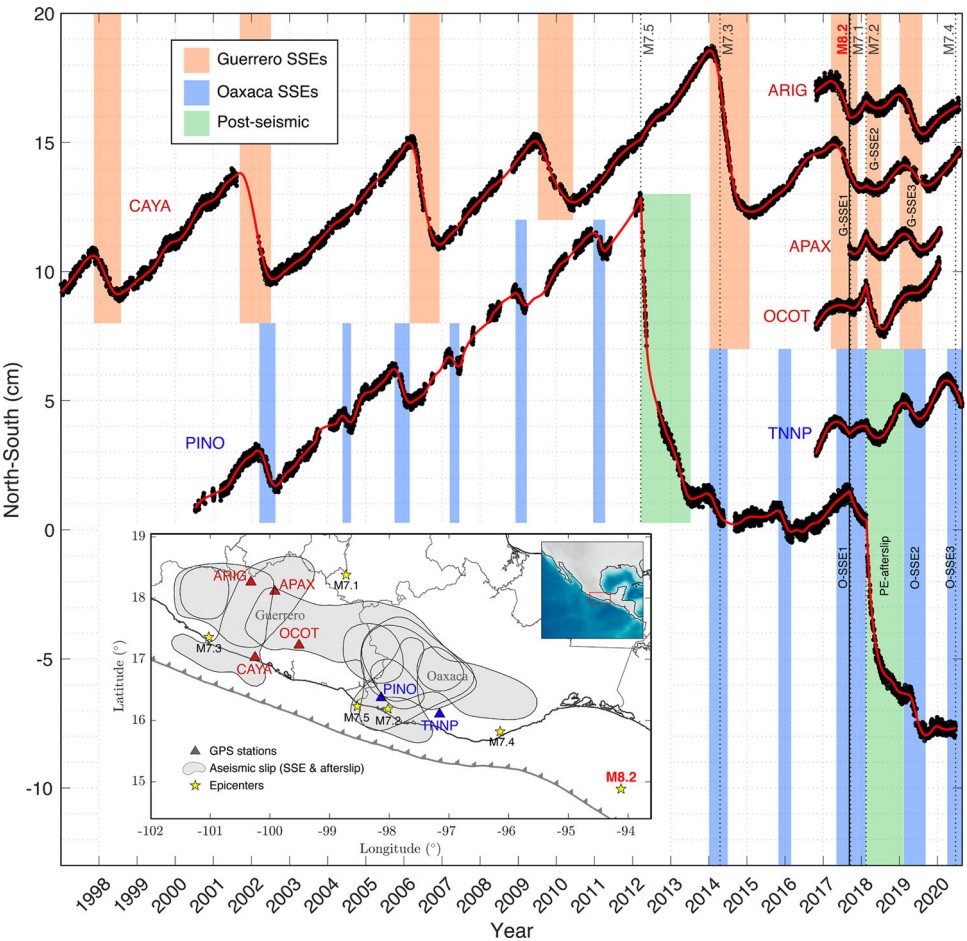

**Fig. 8 Displacement time series in Guerrero (red triangles) and Oaxaca (blue triangles) GPS stations.** The map shows the epicenters of major thrust earthquakes that occurred in the last 23 years in Mexico (M > 7, yellow stars near the coast) and the intraslab Mw7.1 Puebla–Morelos event. Gray shapes delineate the areas of aseismic slip larger than 1 cm determined in this study between November 2016 and October 2019 (see Fig. 3). All the aseismic events (SSEs and afterslip) observed in the time series since 1997 are indicated with vertical-colored bands. Note the change in the temporal deformation patterns throughout the entire region after the great Mw8.2 Tehuantepec earthquake. The O-SSE3, not studied here, initiated two months before the Mw7.4 Huatulco earthquake of June 23, 2020[32].

are exactly those documented for Oaxaca in the literature[23]. M7+ events similar to the Pinotepa earthquake had occurred in Oaxaca and Guerrero[5,23], but none were followed by an SSE during their post-seismic relaxation (i.e., only nine months later). After the Mw7.5 Ometepec earthquake of March 2012, for example, the next SSE took place almost two years later and once the inter-seismic deformation regime had already recovered (Fig. 8). All these observations strongly suggest that, in addition to the dynamic effect of the seismic waves from the Tehuantepec and Pinotepa earthquakes on the dynamics of the ongoing SSEs, the elastic and frictional properties of the plate interface across the entire Mexican subduction zone underwent a change due to the extremely large, unprecedented ground shaking on September 8, 2017.

When seismic waves exceed a certain strain threshold, fault gouge materials undergo abnormal non-linear elastic changes that can bring them to a metastable state facilitating the initiation of earthquakes and SSEs[33,35,37]. Transient changes in both the elastic properties of the crust and the regional seismicity rate have been observed after local and regional earthquakes[33,43]. The greater the damage in the fault core and the lower its effective pressure (e.g., in presence of overpressured fluids), then the non-linear effects of seismic waves will produce a greater drop of the elastic modulus of granular fault gouges (i.e., a material modulus

softening triggered from a lower strain threshold) assisting the unstable interplate slip initiation[34,35]. Although these effects have not yet been observed at the scale of the fault-zone in subduction zones, large seismic waves can affect the continental crust down to its root for several years[44]. It is thus reasonable that the Mw8.2 Tehuantepec earthquake is responsible for the extraordinary disruption of the SSE cycle observed at the regional scale, and even for facilitating the dynamic triggering of the SSEs that we report here. The same hypothesis is valid for the Puebla–Morelos and Pinotepa earthquakes, triggered by the 2017 Guerrero and Oaxaca SSEs (G-SSE1 and O-SSE1), respectively, where the loss of rigidity on both seismogenic fault zones could occur on September 8 (2017) assisting their rupture initiation[35].

As recently discussed by an international community of earthquake modelers[45], this anomalous non-linear behavior of fault-gouge materials should have important implications in friction that are not yet incorporated into R&S fault models. A fault constitutive model that integrates the state laws of both the contact surface and the damaged zone volume subject to these non-linear effects under pressurized fluid conditions, could better explain the interaction between different kinds of dynamic slip instabilities (slow and rapid) and even the sudden regional disruption of the SSE cycles, discussed in this study, after the great Tehuantepec earthquake.

Figure 2 and Supplementary Movie 1 clearly show how the interface coupling, PIC, continuously changes over time. Recent laboratory experiments and theoretical fault models strongly suggest that friction is a (very) sensitive function of the interplate slip-rate where SSEs occur[46,47]. Slow-slip dynamic instabilities, therefore, depend on the velocity field discontinuity at the interface, which is zero only where both plates are completely locked (i.e., in seldom cases). Large temporal variations of the blue areas in Fig. 2 imply large changes in the slip-rate (below the plate convergence velocity), which must therefore have significant implications in the stability of the megathrust not only because of their frictional counterparts but also due to the associated stress changes as recently observed in the hypocentral region of the 2020 Mw7.4 Huatulco earthquake in Oaxaca[32]. Continuous monitoring of both the deformation and the seismic properties of the crust is therefore essential to evaluate the possibility of large earthquakes in the future and to have a clearer idea of the temporal evolution of seismic hazard in subduction zones.

## Methods

**Elastostatic adjoint inversion.** The method used to invert the GPS time series, ELADIN (ELastostatic ADjoint INversion)[29], simultaneously determines the distribution of coupling and SSEs in the plate interface to explain the surface displacements. To this purpose, the method solves a constrained optimization problem based on the adjoint elastostatic equations with Tikhonov regularization terms, a von Karman autocorrelation function, and a Gradient Projection method to guarantee physically-consistent slip restrictions. The main parameters governing the inversions are the correlation length of the von Karman function, $L$, which controls the wavenumber content of the solution, and the precision matrix, which weights the data according to its confidence. We assumed a von Karman Hurst exponent of 0.75 and $L = 40$ km. Comprehensive resolution tests show that, given the problem geometry (i.e., the 3D plate interface and the available stations, Fig. 1), these values maximize the restitution index for slip patches larger than ~80 km length and minimize the data misfit error in the whole plate interface[29].

Although GPS data has been carefully processed to generate the displacement time series (see next section), there are always trailing errors and physical signals that do not correspond to tectonic processes (Supplementary Fig. 1). The precision matrix allows to minimize the effect of such noise in the inversion results and corresponds to the inverse of the data variance per station and time window. To do this, especially in the vertical component, numerous synthetic and real data inversions lead us to adjust the precision matrix (i.e., the data weights) to ensure that, at least, polarities of the vertical-displacement are well explained by the inverted models, while maintaining the best horizontal-displacement fits[29]. The data variance for each component and time window is computed from the differences between daily displacement values and a moving, locally weighted LOESS function (i.e., 2nd order polynomial regressions with half-window time support).

For the inversions, we removed the coseismic displacements produced by the three large earthquakes and improved the 3D plate interface geometry introduced by Radiguet et al.[5] based on the work of Ferrari et al.[48], which compiles relocated seismicity, receiver functions, and tomography studies in southern Mexico. We refined the final geometry beneath Oaxaca based on recent magneto-telluric and receiver function analysis[49,50] (Fig. 1) and assumed a suitable 1D four-layer regional structure[51]. The slip vector is decomposed in the plate-convergence (pc) and pc-perpendicular directions, which vary along the plate interface[52]. Restrictions were imposed to meet reasonable plate coupling constraints (i.e., backslip smaller than the cumulative plate motion in the associated time window) and moderate pc-perpendicular slip by means of an iterative Gradient Projection method[29] so that the slip rake could only vary 30 degrees with respect to the plate convergence direction.

**GPS data processing.** We used continuous records in 57 permanent GPS stations spread across central Mexico (Fig. 1). The stations belong to three different networks: the Mexico-Japan SATREPS-UNAM project[28], the National Seismological Service (SSN-UNAM), and Tlalocnet[53]. GPS data were processed using two different methods: Gipsy 6.4[54] and Gamit/Globk 10.7[55]. For the period between October 23 (2016) to November 22 (2018), after carefully comparing both displacement time series in all stations, we selected those with better signal-to-noise ratio and consistency with nearby stations (Supplementary Fig. 1A). For the period from November 22 (2018) to October 8 (2019), we only selected the selected time series calculated using Gipsy 6.4 (Supplementary Fig. 1B).

The GIPSY displacement time series are estimated with a Precise Point Positioning strategy. The station positions are defined in the International Terrestrial Reference Frame, the year 2014 (ITRF 2014). For daily processing, we used the Jet Propulsion Laboratory final and non-fiducial products (orbits and clocks). We generated observables using 2 model categories: (1) Earth models and

(2) observation models. The Earth models include tidal effects (i.e., solid tides, ocean loading, and tide created by polar motion), Earth rotation (UT1), polar motion, mutation, and precession. Observation models, on the other hand, are related to phase center offsets, tropospheric effects, and timing errors (i.e., relativistic effects). The troposphere delay is estimated like a random walk process. This effect is broken into wet and dry components. The azimuthal gradient and the dry component are estimated using the GPT2 model and mapping function (TGIPSY1). The antenna phase center variations are considered through antenna calibration files. For receiver antennas, the correction is estimated by taking the International GNSS Service (IGS) Antex file. We also applied a wide-lane phase bias to account for the ambiguity resolution and removed outliers.

The GAMIT displacement time series are estimated using a double-difference method that calculates the between-station and satellite differences. It reduces satellite clock and orbit errors, localized atmospheric errors, and cancels the effects of variations in the receiver clocks. The software incorporates final IGS (International GNSS Service) combination solutions for orbits (with accuracies of 1–2 cm) and Earth Orientation Parameters (EOP). Ionospheric and atmospheric corrections were applied during processing. Hydrostatic and water vapor delay are corrected using Vienna Mapping Functions (VMF). Solid Earth tide model (IERS03), ocean tidal loading (FES2004), tables for earth rotation values (nutation IAU2000, polar motion, universal time), and precession constant IAU76 are applied. The resulting GPS time series are calculated in the ITRF 2014 reference frame and then rotated with respect to the fixed North American plate using the rotation pole. Post-processing of daily position time series includes offsetting corrections and outlier removal that was performed with the help of a python-based PYACS package developed by J.-M. Nocquet. Despite integrating all these considerations in the GPS data processing, it is important to notice that the remaining noise may be significant, as it has been recently analyzed in great detail in Guerrero[27].

**Template-matching seismicity analysis.** To detect unreported seismicity within the Mw7.2 Pinotepa earthquake hypocentral region previous to the event, we applied two independent and complementary template matching (TM) techniques. In both cases, the waveform templates were earthquakes reported by the SSN with foci within 30 km from the Pinotepa earthquake hypocenter (Lat: 16.218°, Lon: −98.014°, 16 km depth). We used continuous velocity records in three broadband stations with an epicentral distance smaller than 115 km during a one-year period preceding the earthquake, from March 1, 2017, to February 16, 2018, 23:39 (UTC time of the Mw7.2 earthquake).

The first technique[56] considers three permanent stations (PNIG, YOIG, TXIG) from the SSN network located in the state of Oaxaca (Supplementary Fig. 5A). We used a set of 394 events (templates) (previously identified as repeating earthquakes) reported in the SSN catalog and applied a bandpass Butterworth filter with corner frequencies of 1–8 Hz to reduce the noise, and to remove undesired regional and teleseismic events. For each template, we selected a cross-correlation window starting 1 s before the arrival of the S-wave and ending 5 s after, only one detection is allowed every 25 s (approximately the time needed for the P and S waves of an event to be recorded at all three stations, see Supplementary Fig. 5C) to avoid duplicates of the same event. A detection was confirmed when the stacked correlation coefficient (scc) in the three stations (nine channels) was larger than 0.41 and the median average deviation larger than 25 (Supplementary Fig. 5C). These two values guarantee the best trade-off, with the highest number of detections and the lowest number of false positives. To this end, we performed a grid search in a plane of 4.5 km×4.5 km around each template location (Supplementary Fig. 5A) and looked for the maximum spatial correlation coefficient value. For preventing detectability variations, we only processed those days with data for all components in the three stations.

The second technique considers only the waveforms on the three channels of the station PNIG, the closest site to the earthquake epicenter (21 km, Supplementary Fig. 5B). For generating the templates, we selected 4105 events from the catalog reported by the SSN in the period between March 1, 2017 and March 31, 2018. The waveforms were cut 0.2 s before the P-phase arrival and 0.5 s after the S-phase arrival and filtered using a zero-phase Butterworth bandpass filter with corner frequencies at 3 Hz and 12 Hz. The template matching was performed using the Python package EQcorrscan[57] and the detection threshold was set to 0.9 of the average cross-correlation value in the three channels, which guarantees not only that the detections come from the same place as the templates, but also that our local catalog does not include any false-positives. Single-station detections have proved to be a powerful tool to find earthquakes that are small and located close to certain stations, but that gets too attenuated to be detected at farther stations given high cross-correlation thresholds[58]. Furthermore, a visual inspection of hundreds of waveforms helped us verify that the timing and the relative amplitudes of the ballistic P and S waves in the three components are very similar to the parent templates, guaranteeing that the detected signals are, indeed, earthquakes that share a common hypocentral location as the template events (Supplementary Fig. 5D). For this second matched filter technique we allow inter-event times to be greater or equal to 10 s, keeping only the best-correlated detections.

To assign a common magnitude to all detections, $M_L$, we determined an attenuation relationship specific to PNIG using the LocMagInv code[58] (Supplementary Fig. 6A). Instead of inverting for the magnitudes, we used the

cataloged magnitudes from the SSN for events with SNR greater or equal to 5 and inverted only for the geometric spread, attenuation, and station correction parameters from horizontal displacement records (mm) (i.e., their arithmetic mean). To obtain the displacements, we integrated velocity records in the bandwidth of 3–12 Hz. We only used the available horizontal components for each event.

We detected 3156 events with the first technique (Supplementary Fig. 5A) and 5064 with the second (Supplementary Fig. 5B), which represent a 180% and 350% detection increase, respectively, as compared with the 1125 earthquakes reported by the SSN in the same period and within a 30 km hypocentral radius. Detections from both techniques were integrated into a single catalog avoiding duplicate events (Fig. 5C). Supplementary Fig. 6C shows the frequency-magnitude histograms for both, our TM detections and the SSN catalog, where the cutoff completeness $M_L$ magnitudes correspond to 2.1, 2.4, and 3.2, for local detections (method two), regional detections (method one), and the SSE catalog, respectively.

Since TM method one uses nine seismic channels (i.e., the three components of three stations) at a regional scale, its detections very likely correspond to events with hypocentral locations close to those of the templates that lie, all of them, within 30 km from the Pinotepa earthquake hypocenter. Thus, we used these detections for relatively large events to check how well method two, which only considers local records at PNIG (i.e., the three-component), detected earthquakes within such hypocentral vicinity. Supplementary Fig. 6D shows a Venn diagram for all catalogs where we see that 72% of regional detections were also found using only local records.

### Dynamic perturbations at the plate interface
*From strong motion records.* For the Mw8.2 Tehuantepec event, we used radial and vertical displacement records at 25 s period from strong-motion stations in south-central Mexico (Fig. 5A and Supplementary Fig. 8C) to estimate the strain field produced by the Rayleigh waves fundamental mode at depth, and then the associated CFS (apparent friction coefficient of 0.5) over the 3D plate interface in the plate-convergence slip direction (Supplementary Fig. 9A). Values in Fig. 6A at sites without interface below correspond to a horizontal surface at 50 km depth.

To estimate the surface-wave dynamic deformations (and tractions) at depth from observed ground displacements (i.e., double integration of single-station strong motion records) we followed a two-fold procedure: First, we estimated the displacement at depth (i.e., at the plate interface below each site, Fig. 6A) by modulating the field with the associated surface waves eigenfunctions for the chosen period within a four-layer regional model determined from the dispersion of surface waves[51] (Supplementary Fig. 8D). Then, to estimate the whole strain tensor, we computed the horizontal deformations assuming a phase velocity of 3.5 km/s[38], and the vertical deformations by deriving the eigenfunctions in that direction. Although Love waves can also have SSE triggering potential, in the analysis we only considered perturbations from Rayleigh waves, whose amplitudes differ from those of Love waves by less than a factor of two at distances where the O-SSE1 was developing when the Mw8.2 earthquake took place (420–520 km, Supplementary Fig. 7), indicating that the stress perturbations at the interface induced by the two types of waves should not differ significantly. Supplementary Fig. 9 shows, for the Mw8.2 Tehuantepec earthquake, the traction vector and CFS time series on the 3D plate interface along the plate-convergence slip-rate direction and dilation time series below some selected sites.

To validate our procedure, we compared estimated (with our method) synthetic tractions with the exact solution for the Lamb's problem (i.e., for the wavefield excited by a single vertical force on top of a homogenous halfspace) at depth over a horizontal plane (Supplementary Figs. 8A and 8B). The elastic properties of the medium are $\alpha = 5.6$ km/s, $\beta = 3.233$ km/s, $\rho = 2700$ kg/m³, the surface station lies 300 km away from the source and the buried point is 20 km below the station. In this example, tractions were estimated for a 10 s period. However similar, satisfactory results were obtained for different periods and depths.

*From 3D numerical simulations.* To estimate the Mw7.2 Pinotepa earthquake (complete-wavefield) dynamic perturbations at the plate interface we performed a 3D kinematic-source numerical simulation by means of a hp-adaptive discontinuous Galerkin finite-element method (DGCrack)[41]. The domain is discretized with a non-structured tetrahedral mesh considering a 3D crustal velocity model of the Guerrero-Oaxaca subduction zone[59] that incorporates the real topography and bathymetry, as well as the geometry of the plate interface (Supplementary Fig. 10A). The mesh size is $900 \times 380 \times 104$ km in the along-trench, trench-perpendicular, and vertical directions, respectively, with approximately 11 million elements to achieve a numerical accuracy up to 1 Hz. We run DGCrack in 512 cores on the UNAM supercomputer platform Miztli to complete 260 s of numerical simulation spending 12.5 h of total computer elapsed time. To simulate the finite source, we first used the low-wavenumber slip solution of the Pinotepa earthquake estimated by the USGS (Supplementary Fig. 10B-up). Then, we discretized this solution into subfaults of $1 \times 1$ km and add high-wavenumber slip perturbations that are stochastically generated using a von Karman power spectral density (PSD) function to enhance the radiation of high frequencies following the methodology of Pulido et al.[60] (Supplementary Fig. 10B-down). The slip-rate of every subfault follows a regularized Yoffe function and the rupture evolution is described by the spatial distribution of the slip, rise time, rupture velocity, and peak

time (i.e., the time to reach the peak slip-rate in every subfault) (Supplementary Fig. 10C). These kinematic source parameters are heterogeneously distributed by means of a pseudo-dynamic rupture generator that considers the 1-point and 2-point statistics of each source parameter, as well as their spatial interdependency extracted from dynamic rupture simulations. We validate the earthquake simulation by comparing the horizontal geometric mean of the observed and synthetic peak ground velocities (PGV) in different hard-site strong motion stations (Supplementary Fig. 10D).

Since the resolution of the GPS time series does not allow distinguishing whether the Tehuantepec or Puebla-Morelos earthquakes (only eleven days in between them) produced the abrupt change of the crustal deformation pattern observed in Fig. 4A, we also estimated the dynamic perturbations on the plate interface due to the intraslab, normal-faulting, Mw7.1 Puebla-Morelos event using the same numerical procedure but taking a finite-source solution determined from the inversion of strong motions[17]. Results are shown in Supplementary Fig. 3F, where we appreciate that CFS peak values in the O-SSE1 region (apparent friction coefficient of 0.5) are smaller than those induced by the Tehuantepec earthquake (Fig. 6A) (i.e., <60 kPa). Considering also that the duration of intense shaking by the Mw7.1 is much shorter than that produced by the Mw8.2 Tehuantepec event (i.e., its SSE triggering potential is lower, Supplementary Fig. 11D) and that tremor activity in Oaxaca highly increased a few hours after the Tehuantepec earthquake[26], then we conclude that triggering of the O-SSE1 was produced by seismic waves from the Mw8.2 event.

**Rate and state friction SSE model.** Assuming a 6 cm yr$^{-1}$ plate convergence[52], we developed an R&S fault reference model for the Oaxaca region that spontaneously generates SSEs every 1.5 years with a maximum slip of ~10 cm (Supplementary Fig. 11C), which is a reasonable approximation of the SSE activity in that province[23]. The model assumes a planar fault dipping 13 degrees in a 2D elastic half-space (Fig. 6A and Supplementary Fig. 11A). Following Wei et al.[42] and based on the SSEs slip distributions (Figs. 2C and 3A), the model consists of a velocity-weakening (VW) fault segment between 20 and 45 km depth where SSEs take place encompassed by stable, velocity-strengthening (VS) layers (Supplementary Fig. 11B). Uniform, dynamic stress perturbations from the 2017 Mw8.2 Tehuantepec earthquake and the 2018 Mw7.2 Pinotepa earthquake were inputted around the middle stage of a spontaneously initiated SSE at all depth with different scaling factors (Fig. 7) to consider the variations and uncertainties of both, the reference model and the CFS estimates throughout the SSE region.

## Data availability
Part of the GPS and strong motion data analyzed in this study are available under some restrictions in the repository of the "Servicio Sismológico Nacional de la UNAM" (http://www.ssn.unam.mx). Broadband seismic data is publicly available in the same repository. The rest of the strong motion records are available in the repository of the "Red Acelerográfica del Instituto de Ingeniería de la UNAM" (www.uis.unam.mx). Part of GPS data in the state of Oaxaca are available in the repository of the "TLALOCNet[53] del Instituto de Geofísica de la UNAM" (http://tlalocnet.udg.mx). The rest of the GPS data in the state of Guerrero are not publicly available until March 2026 due to the restriction policies of the SATREPS-UNAM research project. For more information contact the corresponding author.

## Code availability
Custom computer programs and mathematical algorithms that are deemed central to the conclusions of this study are available on request from the corresponding author.

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

## Acknowledgements

We especially thank Yoshihiro Kaneko and Yajing Liu for fruitful discussions to set the R&S fault model, Mathilde Radiguet for discussions on the SSEs inversion, the Servicio Sismológico Nacional of UNAM for providing GPS, broadband, and strong motion data, and the Instituto de Ingeniería of UNAM for providing strong motion data. This work is partially based on GPS data of TLALOCNet and services provided by the GAGE Facility, operated by UNAVCO, Inc., with support from the National Science Foundation and the National Aeronautics and Space Administration under NSF Cooperative Agreement EAR-1724794. SSE inversions and earthquake numerical simulations were performed in the Gaia and Miztli supercomputing platforms of UNAM. This work was supported by UNAM-PAPIIT grants IN113814 and IG100617, UNAM-DGTIC grant LANCAD-312, JICA-JST SATREPS-UNAM grant 15543611, CONACyT grants 6471, 255308 and PN15-639, US NSF grant 1654416, AMEXCID-SRE, and the Ministry of Civil Protection of the State of Guerrero, Mexico.

## Author contributions

V.M.C.-A. conceived and led the study, performed the GPS inversions, estimated the Mw8.2 dynamic CFSs, and wrote the manuscript. J.T. developed the ELADIN inversion method for the study. C.V. designed the study, estimated the static CFSs, performed the DGCrack simulations, and developed processing and visualization tools. M.W. developed and analyzed the SSE R&S models. R.G.G. and L.A.D. performed and analyzed the template-matching seismicity detections. V.K. and T.N. contributed to the GPS data analysis. S.I.F., J.R., and E.K. processed the GPS data and maintained the SATREPS-UNAM GPS stations. M.A.S. participated in the single-station dynamic CFS validation. V.M.C.-A. and Y.I. are the PIs of the SATREPS-UNAM project. All authors reviewed and edited the manuscript.

## Competing interests

The authors declare no competing interests.
