## [Peer Review File · Nature Communications]

REVIEWER COMMENTS

Reviewer #1 (Remarks to the Author):

Short-Term Interaction between Silent and Devastating Earthquakes in Mexico

V. M. Cruz-Atienza et al.

This study shows that three recent earthquakes in Mexico may be related to SSEs, describing a complex sequence of events interacting with each other on a regional scale via quasistatic and/or dynamic perturbations. The thesis is that interaction is conditioned by the transient memory of Earth materials subject to the significant stressing produced by the seismic waves of the Mw8.2 Tehuantepec earthquake, which intensely perturbed the region.

Overall, this is a very nice contribution to our understanding of static and dynamic interaction of slow slip and warrens publication in *Nature Communications*. Stress transfer to active faults has long been recognized as an important aspect of earthquake occurrence. The authors show that changes of the Coulomb Failure Stress explain rupture sequences and seismicity-rate variations. CFS changes smaller than 50 kPa are often spatially well correlated with triggered seismicity and significantly larger than values associated with triggering slow earthquakes.

I do have some concerns that should be considered.

You state that the strong shaking produced in the seismogenic fault by the great Tehuantepec earthquake, could significantly reduce the intraslab frictional strength and thus assist the Mw7.1 Puebla-Morelos rupture initiation driven by the CFS induced by the aseismic slip at the plate interface. You point out that this is the first evidence that an SSE could initiate a devastating intraslab rupture such as the Puebla-Morelos earthquake.

You only considered perturbations from Rayleigh waves in regards to dynamic triggering while acknowledging that Love waves can trigger as well. Was there some logic behind this decision? It seems somewhat arbitrary as both wave phases can trigger. Indeed, shear motions can produce larger nonlinear effects and potentially larger triggering response. Please address this issue in the text.

You state: When seismic waves exceed a certain strain threshold, fault gouge materials undergo abnormal nonlinear changes that can bring them to a metastable state facilitating the triggering of earthquake and SSEs. Although no mechanical changes have yet been observed in the properties of the plate-interface fault-zone due to strong shaking, large seismic waves can affect the continental crust down to its root for several years. It is thus reasonable that the Mw8.2 Tehuantepec earthquake is responsible for the extraordinary disruption of the SSE cycle observed at the regional scale, and even for facilitating the dynamic triggering of the SSEs that we report here.

Indeed, there are stress focusing effects during strong shaking in weak and damaged materials as observed in laboratory studies that would be difficult to discern from surface measurements, supporting your interpretation e.g., Guyer and Johnson 2009; van den Abeele et al, 2000). Overburden would significantly diminish these effects but effective pressures due to high fluid pressures could logically be low, enhancing the effect in the fault zone at nucleation depths.

You further suggest that continuous monitoring of the regional deformation and seismic properties of the crust is essential to assess the possibility of future large earthquakes and thus to have a clearer picture of the temporal evolution of the seismic hazard in subduction zones.

I agree this is of value (e.g. Brengieur et al. 2008; Wang et al 2019; Delorey et al. 2015). At the same time, due to heterogeneous response of the Earth's crust to nonlinear dynamic forcing, and the limited dimension of the damage zone associated with the fault core, this could be somewhat challenging. For instance, using guided waves as a material probe may be one option. It would potentially be highly useful to devise an approach that could be used to probe the fault zone itself. It may be worth discussing this point in more detail in the discussion despite the fact that such research is beyond the scope of this manuscript.

I suggest expanding your discussion section somewhat further to include an overall discussion of the influence of triggering on slip to make the manuscript more impactful and of broader interest. The system was behaving in a semi-orderly manner with slow slip events occurring regularly before the large earthquake. This large earthquake disrupted slip behaviors as you state. Most models such as rate-state do not account for these effects. Indeed, triggering in general, and its effects on earthquake timing are a topic I encourage you to address. The entire discussion of global triggering that was a topic of much interest in the last decade could be brought into the discussion (e.g., Bufe et al., 2005; Michael 2011; Daub et al., 2016, Parsons and Geist, 2012). This could discussion could be in the context of the critical length scale from rate state for example. I think a paragraph on this point could heighten interest.

Other:

You use GPS for your displacement inversions. I was wondering if InSAR is available for the same time period and if you compared inversion results, since your GPS coverage is dense along the coast but sparse as you move inland?

For your first correlation method, you use CC larger than 0.41 and the median average deviation larger than 25. How and why were these values selected? They seem arbitrary.

Earthquake locations. You state in the SI that 'the good fits of the templates that include both P and S direct waves and the coda of the P waves guarantee that detections come from the same hypocentral locations as the template events.' I agree and disagree and this takes some thought. It is plausible could have s-P times with decent correlation from events in an entirely different direction, based on your modest correlation coefficient. It would be beneficial to locate a statistically significant subset of events to demonstrate your hypothesis.

Does figure 1 contain all of the locations from your new catalog?

Figures 2 and 3. Please add descriptions of panels a, b, c etc. associated with each panel. The captions are confusing as is.

I find the smaller fonts in Figure 2 difficult to read. The figure is informative just make the smaller fonts larger.

I hope a GPS/InSAR specialist reviews the manuscript as well, regarding the GPS data processing and inversion. My knowledge is limited in this area.

References.

Brenguier, Florent & Campillo, Michel & Hadziioannou, Celine & Shapiro, Nikolai & Nadeau, Robert & Larose, Eric. (2008). Postseismic Relaxation Along the San Andreas Fault at Parkfield from Continuous Seismological Observations. *Science*. 321. 1478-1481. 10.1126/science.1160943.

Nonlinear mesoscopic elasticity: the complex behaviour of rocks, soil, concrete, RA Guyer, PA Johnson, John Wiley & Sons (2009).

Daub, E. G., Trugman, D. T., and Johnson, P. A. (2015), Statistical tests on clustered global earthquake synthetic data sets, *J. Geophys. Res. Solid Earth*, 120, 5693– 5716, doi:[10.1002/2014JB011777](https://doi.org/10.1002/2014JB011777).

A. A. Delorey, K. Chao, K. Obara, P. A. Johnson, Cascading elastic perturbation in Japan due to the 2012 Mw8.6 Indian Ocean earthquake. *Science Advances* 1, e1500468 (2015).

K Van Den Abeele, P Johnson, A Sutin, Nonlinear elastic wave spectroscopy (NEWS) techniques to discern material damage, Part I: Nonlinear Wave Modulation Spectroscopy (NWMS), *Research on Nondestructive Evaluation* 12 (1), 17-30 (2000).

Parsons, T., and E. L. Geist (2012), Were global $M \geq 8.3$ earthquake time intervals random between 1900 and 2011?, *Bull. Seismol. Soc. Am.*, **102**(4), 1583–1592, doi:[10.1785/0120110282](https://doi.org/10.1785/0120110282).

Q.-Y. Wang et al., Evidence of Changes of Seismic Properties in the Entire Crust Beneath Japan 8 After the Mw 9.0, 2011 Tohoku-oki Earthquake. *Journal of Geophysical Research: Solid Earth* 9 124, 8924-8941 (2019).

Reviewer #2 (Remarks to the Author):

This paper by Cruz-Atienza et al. outlines a complex history of interplay between SSEs and earthquakes in the Mexican subduction system over a period of several years. Their conclusions, that SSEs and earthquakes are intrinsically linked through a variety of triggering mechanisms is an important contribution to the literature documenting such processes. The methodology is well documented such that it seems to be reproducible for further investigation, and I have no concerns about that aspect, except for some minor queries regarding the template-matching, which simply require clarification, rather than reworking. In what follows, I outline my more significant comments, followed by minor editorial suggestions.

General Comments:

The sequence of events (earthquakes/SSEs) is very complex and makes for an extremely dense read given the paper length restrictions. Unfortunately, I don't really see any obvious way around this, other than making sure that all regions/SSEs/earthquakes are labelled with names/regions/magnitudes throughout in both the text and the figures especially to help guide those readers not familiar with the geography. Introducing a coding system, such as labelling SSEs as #1, 2, 3 etc or G1, O2 for Guerrero vs Oaxaca may help too. The summary at the start of the Discussion is nice though, and helps reiterate the sequence.

More specific comments:

Page 6, Lines 12–23: I'm confused by the discussion that the Guerrero SSE helped trigger the Puebla-Morelos earthquake. Did the SSE increase or decrease the CFS on the mainshock fault? From Figure S3 E, it appears the SSE (green line) actually decreased the CFS on the fault, yet the fault-sections on the left show red colours indicating an increase. Are those sections plotting the total CFS change (including PIC + SSE) or just the CFS from the SSE? Perhaps I misunderstood (in which case the comments below can be discarded), but I'd like it if what's being plotted in each case could be clarified more clearly.

If the SSE decreased the CFS on the fault then I find the argument that it triggered the earthquake difficult to reconcile. As I understand, the PIC imposed CFS would continue to increase over the time period, were it not for the SSE disrupting the interface coupling, meaning the total CFS (red and blue lines) on the fault would have been much higher at the time of the M7.1 then they were following the SSE. If anything, the SSE delayed the earthquake, or else triggered it in a way not related to CFS – no? The increase in total CFS referred to on line 15 is also really less than 50 kPa because it's the difference between the lowest Total field point (red line, ~10 kPa) and the interpolated red line value at the M7.1 (~45kPa), so only about 35kPa. I'm not sure where the 50 kPa value in the text comes from.

You also mention at the end of the page that the Tehuantepec EQ was the real kick to the system to trigger the earthquake, so stating that the 'SSE could initiate' the intraslab earthquake seem a bit of a stretch. There's no clear evidence to me that the SSE alone would have triggered the M7.1 without the M8.2 occurring.

Can you also postulate on why the Guerrero SSE stopped following the M7.1, given the increase in CFS on the interface imposed by the EQ (Figure S3F)?

Page 17 Line 17: 25 seconds seems like a long inter-event time restriction, but I'm not sure what the record length is at the recording stations. One of the benefits of TM methods is that they can find overlapping waveforms, which this long inter-event time will minimise. That window should really only be slightly longer than your hypocentral time uncertainty, so you're not duplicating on detections, but still detecting overlapping events.

Related to that, I'd be interested to see how/if the catalogue completeness is time-varying to assess the reliability of the seismicity increase following the M8.2 in Figure 4. Presumably noise levels from

the early aftershock sequence may contribute to decreased detections?

Page 18, Line 7: What do you mean by '0.9 of the average cross-correlation value in the three channels'? and how was the threshold of 0.41 decided on for the first method?

+ later on line 15: 'keeping only the best correlated detection' – was this done subjectively? Or what were the criteria?

Figures:

Figure 1: Can the earthquake names please be labelled alongside the magnitudes to help link in with the text (which often doesn't mention magnitudes).

Figure 2: Labelling of some information is hard to read here, especially the dates along the bottom and magnitudes of the labelled events. I would suggest increasing font size and making sure the earthquake magnitude labels aren't obscured by the coastline. Same comment applies to Figure S3, which has incredibly small text on the left hand panels. I'd also like the SSEs/earthquakes/afterslip and regions to be labelled more clearly to help identification from the text, for those not familiar with the geography. I found I had to work hard to match the text around page 4 lines 12–18 to the figures.

Figure 3: Could the SSEs please be labelled on the time series on the left and right panels, to make clear which N-S direction is slow slip, and which is locking.

Figure 5: The black CFS labels don't stand out well from the dark sea blue colour, at least not when printed. Suggest adding background box to that scale bar.

Figure S3: The text on the left hand panels is really tiny - please enlarge. Also see above comments to make clear whether it's the Total CFS or the SSE CFS that's being plotted in these.

Figure S4: Could the time-line plots like underneath each panel in Figure 2 be added to these plots too?

Minor Editorial Comments:

Page 3, Line 5: Losses 'of' not losses 'for'

Page 3, Line 9: '600 were' not '600 where'

Page 3, Line 17: 'As we discuss' rather than 'as we shall see'

Page 3, Line 19: I would maybe tone down the story-telling language here, as fascinating as it is!

Figure S1: 'Vertical' label on right panel is missing 'r'

Page 5, Lines 3--5: I would add the magnitude of the earthquake here, to help us remember which one is which (only the magnitudes, not the names are labelled on the figures). Or label the earthquakes on the map. Whichever (or both).

Page 5, Line 4: lose 'basically'

Page 5 Line 10: 'at the moment of' : the resolution of this is limited by the GPS sampling, so I wouldn't imply an instantaneous triggering. I would rephrase to 'within a day/a few days of' – if sampling daily solutions.

Page 6, Line 8: suggest change to 'CAN explain rupture sequences and seismicity rate variations remarkably well'.

Page 6 line 13, should this reference Figure 2B, not A?

Page 6, Line 20, 'later' not 'latter'.

Page 8, Line 9: 'cannot be resolved by our GPS' rather than 'escape to our GPS'

Page 18, Line 4: 'were' not 'where'

Reviewer #3 (Remarks to the Author):

Comments to Authors:

Review comments for NCOMMS-20-33002

The manuscript by Cruz-Atienza et al. presents a description of the interactions between SSEs and earthquakes along the Mexican subduction zone during an extraordinary sequence that started with

the Mw8.2 Tehuantepec earthquake in September 2017. The authors demonstrate a complex evolution of the plate interface conditions following the Mw8.2 Tehuantepec earthquake which resulted in both earthquakes and SSEs events triggering. In general, the paper is well written, and the work is clearly presented even if the story is somewhat complex. Most of the results are convincing and indeed show a complex interaction between seismic and aseismic slip partitioning along the Mexican subduction zone. I believe such work is important for the scientific community as it highlights the potential complex interactions and partitioning of slip in subduction zones. I only have a limited number of comments which could be addresses within a moderate revision before the paper can be published in nature communication.

Main Comments:

1. My first and main comment comes from the absence of description of any tremor activities. Are there any triggered tremors by the earthquake sequence? Mexico is known for its ambient and triggered tremor activities (e.g., Payero et al., 2008; Kostoglodov et al., 2010; Zigone et al., 2012; Rivet et al., 2014; Husker et al., 2012; 2019; Frank et al., 2015; 2016). The paper would strongly benefit from the analysis of those tremors. In particular regarding of the triggering of SSEs by both the Tehuantepec and the Pinotepa earthquakes. The location of those tremors (both ambient and triggered) could also highlight the changes in the slip characteristics of the Oaxaca SSE. In particular the westward propagation of the Oaxaca SSE toward Guerrero after the Pinotepa earthquake will be interesting to follow. I encourage the authors to at least include a plot with the detected tremors during the whole period of interest. If possible, a more complete description of the occurrence of ambient and triggered tremors will be useful to have a complete picture of the ongoing processes and interactions.

2. My second comment concerns the general story. Because of the complexity and large number of events, the timeline and the proposed interactions between the elements of the cascade of events are hard to follow. I suggest the authors to add a summary figure which will recap the timing of each event and the proposed interactions (triggering, etc.). This will help the reader to understand the big picture and which interaction (static/dynamic triggering, etc.) have produced with effects along the Mexican subduction zone.

Other comments:

Page 6 line 13: change 'Fig. 2A' by 'Fig. 2B'.

Page 7 line 1: add 'help' before 'initiate'

Page 7 line 6: 'Around the hypocentral region there is a clear rise of CFS...' There is definitively a rise of CFS around the hypocenter of the M7.2 earthquake. However, most of the rise occurs west of Oaxaca in the region that will slip aseismically after the earthquake. I may have missed it, but I don't think this is properly stated in the text. Is there any changes in the seismicity/tremors associated to this rise in CFS?

Page 14 line 11: why 0.75 and L=40km. Which comprehensive resolution tests did you performed? In general, it would be great to add a sentence about all the performed tests and where to find the results. This may come from the fact that the methodological is curenly under review and therefore not published yet.

Page 15 line 18: The GPS time series are restricted to October 23, 2016 to November 22, 2018. I would be helpful to add longer example time series for a typical station in Oaxaca and Guerrero. This

could be added in supplementary material. Such figure will also highlight the unusual analyzed sequence compare to what is typical observed in Mexico.

Page 16 line 6: replace 'as' by 'a' before 'random'.

Page 17 line 7: why 30km around the Pinotepa Eq and not 50km or 20km?

Page 17 line 13: How were the 394 events selected? Is it the whole catalogue? Only a selection? If it's a selection what are the selection criteria?

Page 20 line 6: replace 'stain' by 'strain'.

Figure 3: It looks like there is some perturbations in the time series of the spontaneous SSE (right on the Figure) associated to the earthquakes (both the M8.2 and M7.2 events). Is there really something there?

Figure 3A: can you add some GPS stations in Guerrero? I'll be curious to see the evolution of the first SSE in Guerrero even if it occurs mainly before the M8.2 earthquake.

Figure 3B: On the map there is no GPS stations plotted east of the 7.2 earthquake. If stations are available, it will be interesting to include them to investigate the eastward aseismic slip.

Figure 5: It's very hard to distinguish the black and green lines. Please change the green color in red or something else.

Best Regards

Reviewer #1 (Remarks to the Author):

Short-Term Interaction between Silent and Devastating Earthquakes in Mexico

V. M. Cruz-Atienza et al.

This study shows that three recent earthquakes in Mexico may be related to SSEs, describing a complex sequence of events interacting with each other on a regional scale via quasistatic and/or dynamic perturbations. The thesis is that interaction is conditioned by the transient memory of Earth materials subject to the significant stressing produced by the seismic waves of the Mw8.2 Tehuantepec earthquake, which intensely perturbed the region.

Overall, this is a very nice contribution to our understanding of static and dynamic interaction of slow slip and warrens publication in *Nature Communications*. Stress transfer to active faults has long been recognized as a important aspect of earthquake occurrence. The authors show that changes of the Coulomb Failure Stress explain rupture sequences and seismicity-rate variations. CFS changes smaller than 50 kPa are often spatially well correlated with triggered seismicity and significantly larger than values associated with triggering slow earthquakes.

- We are grateful for and appreciate the reviewer's general comments on our work.

I do have some concerns that should be considered. You state that the strong shaking produced in the seismogenic fault by the great Tehuantepec earthquake, could significantly reduce the intraslab frictional strength and thus assist the Mw7.1 Puebla-Morelos rupture initiation driven by the CFS induced by the aseismic slip at the plate interface. You point out that this is the first evidence that an SSE could initiate a devastating intraslab rupture such as the Puebla-Morelos earthquake.

You only considered perturbations from Rayleigh waves in regards to dynamic triggering while acknowledging that Love waves can trigger as well. Was there some logic behind this decision? It seems somewhat arbitrary as both wave phases can trigger. Indeed, shear motions can produce larger nonlinear effects and potentially larger triggering response. Please address this issue in the text.

- Although transient shearing in rocks is produced by both Love and Rayleigh waves, it is true that Love waves could have produced even greater shear deformations than Rayleigh waves. However, considering that our estimated deformations are about an order of magnitude greater than those triggering widespread tremor in Japan due to the seismic waves from the 2004 Sumatra-Andaman earthquake, and the R&S fault model using these Rayleigh-wave perturbations clearly shows that traction waves with those amplitudes should have strong implications in the SSE development, the choice of the type of surface wave should not be so decisive. If

Love waves had a larger amplitude, this would endorse our conclusions. The following figure shows the peak ground displacement (PGD) of the Rayleigh and Love waves of the Mw8.2 earthquake as a function of epicentral distance. The PGDs differ by less than a factor of two at distances where the O-SSE1 was developing at the time of the earthquake, suggesting that the stress perturbations at the interface induced by the two types of waves should not differ significantly. We have pointed this out in the manuscript (Methods).

Figure. Peak ground displacements of Rayleigh and Love waves from the Mw8.2 earthquake as a function of epicentral distance. They differ in less than a factor of two at distances where the O-SSE1 was developing at the moment of the earthquake.

You state: When seismic waves exceed a certain strain threshold, fault gouge materials undergo abnormal nonlinear changes that can bring them to a metastable state facilitating the triggering of earthquake and SSEs. Although no mechanical changes have yet been observed in the properties of the plate-interface fault-zone due to strong shaking, large seismic waves can affect the continental crust down to its root for several years. It is thus reasonable that the Mw8.2 Tehuantepec earthquake is responsible for the extraordinary disruption of the SSE cycle observed at the regional scale, and even for facilitating the dynamic triggering of the SSEs that we report here.

Indeed, there are stress focusing effects during strong shaking in weak and damaged materials as observed in laboratory studies that would be difficult to discern from surface measurements, supporting your interpretation e.g., Guyer and Johnson 2009; van den Abeele et al, 2000). Overburden would significantly diminish these effects but effective pressures due to high fluid pressures could logically be low, enhancing the effect in the fault zone at nucleation depths.

- In addition to the references to Johnson et al. (2005 and 2012) that were already included, we have added others you have suggested in the Discussion section and mentioned the role that both the effective pressure and the damage level of the fault zone have important implications in the deformation threshold for the material to experience the abnormal non-linear effects.

You further suggest that continuous monitoring of the regional deformation and seismic properties of the crust is essential to assess the possibility of future large earthquakes and thus to have a clearer picture of the temporal evolution of the seismic hazard in subduction zones.

I agree this is of value (e.g Brengieur et al. 2008; Wang et al 2019; Delorey et al. 2015). At the same time, due to heterogenous response of the Earth's crust to nonlinear dynamic forcing, and the limited dimension of the damage zone associated with the fault core, this could be somewhat challenging. For instance, using guided waves as a material probe may be one option. It would potentially be highly useful to devise an approach that could be used to probe the fault zone itself. It may be worth discussing this point in more detail in the discussion despite the fact that such research is beyond the scope of this manuscript.

- We fully agree on the importance and implications that trapped (and guided) waves in fault zones may have on the dynamics of slip (whether slow or fast). This has been demonstrated with theoretical models, and seismic energy entrapment has been observed on shallow faults. However, as mentioned, this discussion is beyond the scope of our study so we prefer to allocate the available space to discuss aspects closer to the evidence presented here.

I suggest expanding your discussion section somewhat further to include an overall discussion of the influence of triggering on slip to make the manuscript more impactful and of broader interest. The system was behaving in a semi-orderly manner with slow slip events occurring regularly before the large earthquake. This large earthquake disrupted slip behaviors as you state. Most models such as rate-state do not account for these effects. Indeed, triggering in general, and its effects on earthquake timing are a topic I encourage you to address. The entire discussion of global triggering that was a topic of much interest in the last decade could be brought into the discussion (e.g., Bufe et al., 2005; Michael 2011; Daub et al., 2016, Parsons and Geist, 2012). This could discussion could be in the context of the critical length scale from rate state for example. I think a paragraph on this point could heighten interest.

- We welcome the reviewer's suggestions and fully agree that the points he mentions should be addressed somehow in the discussion. Therefore, we have deepened the explanation of the anomalous non-linear response of damaged materials and its implications in earthquake triggering, including some more references. As for previous observations where transient changes in the seismic properties of the crust have been observed, in addition to the reference to Wang et al. (2019) that we already included, we have added others and notably that of Delorey et al. (2015). We also mention the inability of today's mechanical (theoretical) source models to include these non-linear effects into the frictional fault response and suggest a strategy to do so. We point out that a constitutive model with these characteristics could help to better explain the interaction between the events and the sudden disruption of the SSE cycles at a regional scale from the great Tehuantepec earthquake. Finally, we refer to very recent observations and results that allow us to better understand the implications of the temporal changes of the coupling in the interface that we have found along the sequence, which is critical to assess the dynamic stability of the megathrust. Given the space limitations, we preferred not to discuss either the fault-zone wave-guide issue or the very extensive literature on earthquake triggering to avoid any confusing omissions. We hope that with these modifications the reviewer will feel more satisfied with the final discussion.

Other:

You use GPS for your displacement inversions. I was wondering if InSAR is available for the same time period and if you compared inversion results, since your GPS coverage is dense along the coast but sparse as you move inland?

- This point is very important. Confidence in the data is essential in order to conclude something real. Free InSAR data for the study region exists thanks to the European Union's Sentinel satellite. A complete analysis of these images is being carried out now at ISTERre, France. However, comparisons with our inversion results are not easy since this research group is focused on the estimation of the interplate coupling only, which entails a treatment of the LOS (InSAR) displacements which, in fact, makes a direct comparison with our solutions impossible since we solve simultaneously for the aseismic (relaxing) slip and the coupling. In a very recent article (<https://doi.org/10.1002/essoar.10504796.3>) we have performed the simultaneous inversion of GPS and InSAR data for the 2020 Huatulco earthquake in Oaxaca, finding very consistent displacements between both satellite observations. However, to be sure that our data and subsequent inversions presented here are acceptably good, we have done two things: (1) we compared (validated) our inversion results for the 2006 SSE in Guerrero with two previously published solutions, which can be found in the accompanying article by Tago et al. (2020, under minor revision at GJIInt), where we introduced the ELADIN inversion method; and (2) all GPS data inverted in this work were processed using two independent methods (Gipsy and Gamit) and we verified both solutions (time

series) were always consistent. For these reasons we are confident in the reliability of our solutions. Please see our response to Reviewer 3 below, where we also address this issue and provide some supporting figures.

For your first correlation method, you use CC larger than 0.41 and the median average deviation larger than 25. How and why were these values selected? They seem arbitrary.

- We agree that the selection of these values may seem arbitrary at first glance, nonetheless the choice was the result of a meticulous inspection of the waveforms, the tradeoff between MAD and CC threshold, and the comparison of results between the two TM techniques. MAD measures how many times a detection is possible above the median absolute deviation of the distribution of the cross-channel sum of the maximum cross-correlation values. We started by determining what value of MAD was less prone to false detections and which values better complemented the results obtained by the second technique. We evaluated four sets of detections for MAD values of 15, 20, 25 and 30 and visually inspect the resulting waveforms. Low MAD numbers (M15, M20) led to a substantially high number of false positives. On the other hand, a high MAD=30 drastically reduced the capabilities of the Match Filter barely proving additional information in comparison to the first technique. Next, we evaluated how the selection of the MAD influence the CC coefficient for each set. As we decrease MAD, CC also decreases, increasing the number of false detections. Particularly, we noticed a bending point where CC quickly decreases for the same MAD threshold. Visual inspection showed that lowering CC deteriorates the quality of the detections specially for the detections beyond this point (CC=0.41). Thus, we took this turning point as a minimum CC as shown in the next figure which roughly discarded 10% of original detections. This combination of thresholds provided the best tradeoff between the number of detections as well as the minimum number of false positives. We have clarified this in the text.

Figure. Tradeoff between the number of detections and parameters (MAD and CC). We chose the threshold of $CC=0.41$ as the bending point where the correlation CC quickly decreases for $MAD=25$.

Earthquake locations. You state in the SI that ‘the good fits of the templates that include both P and S direct waves and the coda of the P waves guarantee that detections come from the same hypocentral locations as the template events.’ I agree and disagree and this takes some thought. It is plausible could have s-P times with decent correlation from events in an entirely different direction, based on your modest correlation coefficient. It would be beneficial to locate a statistically significant subset of events to demonstrate your hypothesis.

- If the waveform of the P-wave in all three components is perfectly explained (i.e., its amplitude and polarization), then the azimuth and the incidence angle of the

incoming wave are known. If the arrival of the S-wave is also explained, then we also have a measurement of the hypocentral distance (S-P arrival times). In our single-station TM technique we only keep those detections that have an average correlation coefficient in all three components greater than 0.9 so that, in addition to explaining the direct P and S waveforms very well (see Figure S5D), we also explain the coda of the P wave that represents a part of Green's function between the hypocenter and the station, which is unique to those two points in space. For all these reasons we are convinced that most of our detections occurred in the immediate vicinity of the associated templates. We have been extremely careful to ensure that the local detections correspond to events very close to the parent templates. This is demonstrated in Figures S6C and S6D by comparing single-station and regional (9 channels) larger-magnitude detections and finding a very high consistency.

Does figure 1 contain all of the locations from your new catalog?

- We understand that the reviewer is referring to the SSEs catalog, and the answer is yes. Such catalog can also be found in the new Figure 3A.

Figures 2 and 3. Please add descriptions of panels a, b, c etc. associated with each panel. The captions are confusing as is.

- We have improved the captions of both figures in addition to having included the code names of each of the aseismic events.

I find the smaller fonts in Figure 2 difficult to read. The figure is informative just make the smaller fonts larger.

- Done

I hope a GPS/InSAR specialist reviews the manuscript as well, regarding the GPS data processing and inversion. My knowledge is limited in this area.

- They have

References.

- We have included references to Brenguier et al., Delorey et al. and Van Den Abeele et al. The reference to Wang et al. was already in the manuscript.

Johnson, John Wiley & Sons (2009). Nonlinear mesoscopic elasticity: the complex behaviour of rocks, soil, concrete, RA Guyer, PA

Brenguier, Florent & Campillo, Michel & Hadziioannou, Celine & Shapiro, Nikolai & Nadeau, Robert & Larose, Eric. (2008). Postseismic Relaxation Along the San Andreas

Fault at Parkfield from Continuous Seismological Observations. *Science*. 321. 1478-1481. [10.1126/science.1160943](https://doi.org/10.1126/science.1160943).

Daub, E. G., Trugman, D. T., and Johnson, P. A. (2015), Statistical tests on clustered global earthquake synthetic data sets, *J. Geophys. Res. Solid Earth*, 120, 5693–5716, [doi:10.1002/2014JB011777](https://doi.org/10.1002/2014JB011777).

A. A. Delorey, K. Chao, K. Obara, P. A. Johnson, Cascading elastic perturbation in Japan due to the 2012 Mw8.6 Indian Ocean earthquake. *Science Advances* 1, e1500468 (2015).

K Van Den Abeele, P Johnson, A Sutin, Nonlinear elastic wave spectroscopy (NEWS) techniques to discern material damage, Part I: Nonlinear Wave Modulation Spectroscopy (NWMS), *Research on Nondestructive Evaluation* 12 (1), 17-30 (2000).

Parsons, T., and E. L. Geist (2012), Were global $M \geq 8.3$ earthquake time intervals random between 1900 and 2011?, *Bull. Seismol. Soc. Am.*, 102(4), 1583–1592, [doi:10.1785/0120110282](https://doi.org/10.1785/0120110282).

Q.-Y. Wang et al., Evidence of Changes of Seismic Properties in the Entire Crust Beneath Japan 8 After the Mw 9.0, 2011 Tohoku-oki Earthquake. *Journal of Geophysical Research: Solid Earth* 9 124, 8924-8941 (2019).

Reviewer #2 (Remarks to the Author):

This paper by Cruz-Atienza et al. outlines a complex history of interplay between SSEs and earthquakes in the Mexican subduction system over a period of several years. Their conclusions, that SSEs and earthquakes are intrinsically linked through a variety of triggering mechanisms is an important contribution to the literature documenting such processes. The methodology is well documented such that it seems to be reproducible for further investigation, and I have no concerns about that aspect, except for some minor queries regarding the template-matching, which simply require clarification, rather than reworking. In what follows, I outline my more significant comments, followed by minor editorial suggestions.

- We greatly appreciate the reviewer's general comments and feedback on our work.

General Comments:

The sequence of events (earthquakes/SSEs) is very complex and makes for an extremely dense read given the paper length restrictions. Unfortunately, I don't really see any obvious way around this, other than making sure that all regions/SSEs/earthquakes are labelled with names/regions/magnitudes throughout in both the text and the figures especially to help guide those readers not familiar with the geography. Introducing a coding system, such as labelling SSEs as #1, 2, 3 etc or G1, O2 for Guerrero vs Oaxaca may help too. The summary at the start of the Discussion is nice though, and helps reiterate the sequence.

- We perfectly understand this concern, which was also pointed out by another reviewer. Following the above suggestions, to make it easier for the reader to create a mental map of the succession of events, we have done three things. First, we have coded all aseismic events (e.g. G-SSE1, G-SSE2, O-SSE1,...) across the manuscript and included that terminology in different figures (e.g. Figs. 2, 4 and 6). Then, we have created a movie of the sequence (linearly interpolated every 30 days) where all aseismic events and earthquakes can easily be followed up, and added it as a supplementary material (we strongly suggest the reviewer to watch the movie). Finally, we also included a new figure, Figure 3, where the spatial and temporal evolution of the aseismic slip sequence (and earthquakes) can also be seen at once together with the type of interaction between the events. We have included the appropriate figure references in the main text respecting the events' codes (section "Plate Interface Aseismic Slip History"). We hope this will satisfy the reviewer request.

More specific comments:

Page 6, Lines 12—23: I'm confused by the discussion that the Guerrero SSE helped trigger the Puebla-Morelos earthquake. Did the SSE increase or decrease the CFS on the

mainshock fault? From Figure S3 E, it appears the SSE (green line) actually decreased the CFS on the fault, yet the fault-sections on the left show red colours indicating an increase. Are those sections plotting the total CFS change (including PIC + SSE) or just the CFS from the SSE? Perhaps I misunderstood (in which case the comments below can be discarded), but I'd like it if what's being plotted in each case could be clarified more clearly.

If the SSE decreased the CFS on the fault then I find the argument that it triggered the earthquake difficult to reconcile. As I understand, the PIC imposed CFS would continue to increase over the time period, were it not for the SSE disrupting the interface coupling, meaning the total CFS (red and blue lines) on the fault would have been much higher at the time of the M7.1 then they were following the SSE. If anything, the SSE delayed the earthquake, or else triggered it in a way not related to CFS – no?

- The confusion certainly stems from the fact that we do not explicitly mention in Figures S3A-D that the plotted field is the total CFS (i.e., the sum of PIC and SSE contributions). As the reviewer correctly points out and as stated in the main text and Figure S3E and its caption, the increase in the CFS does not come directly from the SSE but from the neighboring interface region in a coupling regime. One of the most striking results emerging from our inversions is the continuous variation of PIC over the 3.5-year period and how these changes correlate (over time) with the occurrence of stress-release slip processes (i.e., SSE and afterslip). This can be clearly seen in the Supplementary Move 1, for example. Therefore, although most of the positive CFS increments are caused by changes in PIC, as revealed by the blue curve in Figure S3E, our results indicate that these changes would not have occurred if the SSE had not taken place. In this sense, the increase in CFS is an indirect consequence of the SSE. We have made some clarifications in both the Figure S3 caption and the main text, and added also a new reference to a recently submitted work (Villafuerte et al., EPSL, 2020; e.g., their Figure 6) where we study in depth the correlation of such PIC variations with the occurrence of stress-release slip processes (i.e., SSEs).
 - Reference: Villafuerte, C., V. M. Cruz-Atienza, J. Tago, D. Solano, R. Garza-Girón, S. I. Franco, L. A. Domiguez and V. Kostoglodov. Slow slip events and megathrust coupling changes reveal the earthquake potential before the 2020 Mw 7.4 Huatulco, Mexico, event. Under review in Earth and Planetary Science Letters, <https://doi.org/10.1002/essoar.10504796.3>, November, 2020.

The increase in total CFS referred to on line 15 is also really less than 50 kPa because it's the difference between the lowest Total field point (red line, ~10 kPa) and the interpolated red line value at the M7.1 (~45kPa), so only about 35kPa. I'm not sure where the 50 kPa value in the text comes from.

- This is entirely true. The value of 50 kPa was "inherited" from a previous exercise that we did not review in detail. We have corrected the value in the text. Thank you for the observation.

You also mention at the end of the page that the Tehuantepec EQ was the real kick to the system to trigger the earthquake, so stating that the 'SSE could initiate' the intraslab earthquake seem a bit of a stretch. There's no clear evidence to me that the SSE alone would have triggered the M7.1 without the M8.2 occurring.

- This is not exact. What we argue is that the seismic waves of the Tehuantepec earthquake were able to reduce the fault strength in the hypocentral zone of the Puebla-Morelos earthquake in such a way that the increase of CFS, induced by the interface slip processes, reached the rupture threshold before. In that sense, those processes are responsible for triggering the earthquake, not the Tehuantepec event. It is impossible to know whether the Puebla-Morelos earthquake would have happened later if the 8.2 earthquake had not occurred eleven days earlier. We simply postulate that mechanism (i.e., the reduction of fault strength with the passage of waves) as a plausible hypothesis given the observed regional disturbance of the deformation processes. We have tried to make this point clearer in the text.

Can you also postulate on why the Guerrero SSE stopped following the M7.1, given the increase in CFS on the interface imposed by the EQ (Figure S3F)?

- The static CFS on the plate interface imparted by the Puebla-Morelos earthquake are not those shown in Figure S3F, as suggested by the reviewer, but the peak dynamic CFS values produced by its seismic waves (see the clarification in the Figure caption). Figure 6B shows that the static stresses promote the arrest of the SSE only on its very north segment. This is why we have investigated in depth the question raised by the reviewer before our first submission by means of the rate and state SSE model subject to the dynamic perturbations produced by the Puebla-Morelos event on the plate interface. We wanted to assess whether these waves were capable of halting the ongoing SSE. Our conclusion was that they can do so by accelerating the release of stress (i.e., the slip) during the last stage of the SSE to run out of elastic energy sooner and then stop. However, since our GPS inversions do not have adequate temporal resolution, we could not identify any late slip acceleration that would validate the model predictions. This is why we decided to leave this analysis out of the manuscript.

Page 17 Line 17: 25 seconds seems like a long inter-event time restriction, but I'm not sure what the record length is at the recording stations. One of the benefits of TM methods is that they can find overlapping waveforms, which this long inter-event time will minimise. That window should really only be slightly longer than your hypocentral time uncertainty, so you're not duplicating on detections, but still detecting overlapping events.

- We understand this point. However, multiple tests varying the inter-event window (even allowing their overlap) led us to conclude that the optimal way to avoid false detections is to use the 25 s window, which corresponds approximately to the time needed for the P and S waves of an event to be recorded at all three stations (see Figure S5C). Although for this reason our regional catalog might underestimate the number of events, this should not have any major implication in our final results since most of the detections were made with the second method from the closest station (Figs. S5B and S6). Additionally, we are mostly interested in the detection of foreshocks rather than aftershock, when overlapping between events is by far a less common issue. Thus, we set this parameter to a relatively high value to avoid possible false detections, we noticed that reducing this value made false detections more likely to occur. We have clarified this in the text.

Related to that, I'd be interested to see how/if the catalogue completeness is time-varying to assess the reliability of the seismicity increase following the M8.2 in Figure 4. Presumably noise levels from the early aftershock sequence may contribute to decreased detections?

- We appreciate the reviewer's concern on this issue, as it is of great importance to guarantee the reliability of the temporal changes in seismicity rate. The magnitude of completeness of 2.1 that we impose on the catalog is based on the maximum peak of the frequency-magnitude distribution of the template-matched catalog (Figure S6C). Indeed, this distribution reflects the completeness of the catalog as a whole, rather than the possible temporal variations in completeness that, surely, exists. We computed the temporal variations in M_c using ZMAP software to test these variations as suggested. However, temporal analysis of the magnitudes (figure below (a)) shows that, our completeness value (red line) is robust and even conservative since variations of seismicity are seen above that completeness level. From this figure one can appreciate that a magnitude of completeness of 2.1 is appropriate for our catalog.

Figure. Completeness (a) and magnitude (b) of our template matching seismic catalog as a function of time. Data gaps are indicated with gray bars.

Page 18, Line 7: What do you mean by ‘0.9 of the average cross-correlation value in the three channels’? and how was the threshold of 0.41 decided on for the first method?

- We address here only the selection of the threshold (CC=0.9) for the second technique since the threshold for the first technique (CC=0.41) has already been explained to Reviewer 1 (above). As we are interested in the local variability of the seismicity rate in the hypocentral region, our second technique focuses on using a single, broadband, 3 channels station (PNIG), which is the closest to the source. For this technique we used an intuitive thresholding method: the mean of the cross-correlations for each one of the three channels has to be ≥ 0.9 (i.e. $\text{sum}(\text{CC}_{v,e,n})/3 \geq 0.9$). We selected 0.9 after carefully revising the detections in intervals between 0.99 down to 0.8 in 0.01 increments. By randomly choosing a set of detections in each interval, we came to the conclusion that the value of 0.9 guarantees not only that the detections come from the same location as the templates, but also that the results from our local detection technique do not include any false-positives. We have clarified this in the text.

Later on line 15: ‘keeping only the best correlated detection’ – was this done subjectively? Or what were the criteria?

- There are some detections that can be found by different templates. The matches do not happen exactly at the same time (i.e., there are slight differences in the maximum CC lag time) and they have different CC values. In order to avoid counting these detections more than once, we group all of our matches in windows of 10 seconds and only keep the one with the highest cross-correlation value. The choice of a 10s window is arbitrary; however, it is long enough to avoid having artifacts from counting the same detection twice.

Figures:

Figure 1: Can the earthquake names please be labelled alongside the magnitudes to help link in with the text (which often doesn’t mention magnitudes).

- Done

Figure 2: Labelling of some information is hard to read here, especially the dates along the bottom and magnitudes of the labelled events. I would suggest increasing font size and making sure the earthquake magnitude labels aren’t obscured by the coastline. Same comment applies to Figure S3, which has incredibly small text on the left hand panels. I’d also like the SSEs/earthquakes/afterslip and regions to be labelled more clearly to help identification from the text, for those not familiar with the geography. I found I had to work hard to match the text around page 4 lines 12–18 to the figures.

- We have labeled all aseismic events in Figures 2, 4 and 6 following the coded names introduced in the main text (page 4). We have also increased the size of the fonts beside other display improvements. The coded names are now systematically respected throughout the manuscript and figures, which makes the whole story much more accessible to the reader. We have also included de new Figure 3 with new graphical elements for the same purpose.

Figure 3: Could the SSEs please be labelled on the time series on the left and right panels, to make clear which N-S direction is slow slip, and which is locking.

- Done

Figure 5: The black CFS labels don't stand out well from the dark sea blue colour, at least not when printed. Suggest adding background box to that scale bar.

- Done

Figure S3: The text on the left hand panels is really tiny - please enlarge. Also see above comments to make clear whether it's the Total CFS or the SSE CFS that's being plotted in these.

- Done

Figure S4: Could the time-line plots like underneath each panel in Figure 2 be added to these plots too?

- We understand that it would be easier to interpret the figure with the timelines under each panel. However, this would mean reducing the size of each panel very much. We believe that as it stands, the figure can be read with the necessary clarity. We hope that the reviewer will grant this request.

Minor Editorial Comments:

- All the following comments have been addressed in the manuscript.

Page 3, Line 5: Losses 'of' not losses 'for'

Page 3, Line 9: '600 were' not '600 where'

Page 3, Line 17: 'As we discuss' rather than 'as we shall see'

Page 3, Line 19: I would maybe tone down the story-telling language here, as fascinating as it is!

Figure S1: 'Vertical' label on right panel is missing 'r'

Page 5, Lines 3--5: I would add the magnitude of the earthquake here, to help us remember which one is which (only the magnitudes, not the names are labelled on the figures). Or label the earthquakes on the map. Whichever (or both).

Page 5, Line 4: lose 'basically'

Page 5 Line 10: 'at the moment of': the resolution of this is limited by the GPS sampling, so I wouldn't imply an instantaneous triggering. I would rephrase to 'within a day/a few days of' – if sampling daily solutions.

Page 6, Line 8: suggest change to 'CAN explain rupture sequences and seismicity rate variations remarkably well'.

Page 6 line 13, should this reference Figure 2B, not A?

Page 6, Line 20, 'later' not 'latter'.

Page 8, Line 9: 'cannot be resolved by our GPS' rather than 'escape to our GPS'

Page 18, Line 4: 'were' not 'where'

Reviewer #3 (Remarks to the Author):

Comments to Authors:

Review comments for NCOMMS-20-33002

The manuscript by Cruz-Atienza et al. presents a description of the interactions between SSEs and earthquakes along the Mexican subduction zone during an extraordinary sequence that started with the Mw8.2 Tehuantepec earthquake in September 2017. The authors demonstrate a complex evolution of the plate interface conditions following the Mw8.2 Tehuantepec earthquake which resulted in both earthquakes and SSEs events triggering. In general, the paper is well written, and the work is clearly presented even if the story is somewhat complex. Most of the results are convincing and indeed show a complex interaction between seismic and aseismic slip partitioning along the Mexican subduction zone. I believe such work is important for the scientific community as it highlights the potential complex interactions and partitioning of slip in subduction zones. I only have a limited number of comments which could be addresses within a moderate revision before the paper can be published in nature communication.

- We are grateful for the assessment of our work by the reviewer as well as for the observations made.

Main Comments:

1. My first and main comment comes from the absence of description of any tremor activities. Are there any triggered tremors by the earthquake sequence? Mexico is known for its ambient and triggered tremor activities (e.g., Payero et al., 2008; Kostoglodov et al., 2010; Zigone et al., 2012; Rivet et al., 2014; Husker et al., 2012; 2019; Frank et al., 2015; 2016). The paper would strongly benefit from the analysis of those tremors. In particular regarding of the triggering of SSEs by both the Tehuantepec and the Pinotepa earthquakes. The location of those tremors (both ambient and triggered) could also highlight the changes in the slip characteristics of the Oaxaca SSE. In particular the westward propagation of the Oaxaca SSE toward Guerrero after the Pinotepa earthquake will be interesting to follow. I encourage the authors to at least include a plot with the detected tremors during the whole period of interest. If possible, a more complete description of the occurrence of ambient and triggered tremors will be useful to have a complete picture of the ongoing processes and interactions.

- We fully agree with this concern. In fact, tectonic tremor in Mexico has served to highlight and better understand the behavior and physics driving the aseismic slip of the plate interface. In that sense, our research team has made significant efforts that we do not ignore (e.g., Husker et al., 2012; Cruz-Atienza et al., 2015; Villafuerte and Cruz-Atienza, 2018; Maury et al., 2016, 2018; Cruz-Atienza et al., 2018). However, during the particular period studied in this work, there are at least two previous publications (referred to in the manuscript) that demonstrate the increase

in tremor activity in Guerrero, Oaxaca (Husker et al., 2019; figure below) and Jalisco (Miyazawa and Santoyo, 2020; figure below) following the M8.2 and M7.2 earthquakes, which is consistent with our geodetic observations. Considering these works that clearly show (specially Husker et al., 2019) the rise in tremor activity in Oaxaca at the very same moments we detected the SSEs (i.e. at the time of both earthquakes), but also the amount of information (results and methods), complexity and strict space limitations of the present work (we already reached the limit of figures allowed in both the main text and supplement), we consider that including some complementary tremor analysis (and related figures) will not provide further evidence of the interaction we address between the events, and would mean an important restructuring of the text that could obscure the analysis and reasoning. We are very grateful, however, for the reviewer's observation.

References:

- Cruz-Atienza, V. M., A. Husker, D. Legrand, E. Caballero and V. Kostoglodov. Non-Volcanic Tremor Locations and Mechanisms in Guerrero, Mexico, from Energy-based and Particle-Motion Polarization Analysis. *Journal of Geophysical Research*, 120, doi:10.1002/2014JB011389, 2015.
- Cruz-Atienza, V. M., C. D. Villafuerte and H. S. Bhat. Rapid tremor migration and pore-pressure waves in subduction zones. *Nature Communications*, doi:10.1038/s41467-018-05150-3, 2018.
- Husker, A. L., V. Kostoglodov, V. M. Cruz-Atienza, D. Legrand, N. Shapiro, J. S. Payero and M. Campillo. Temporal variations of non-volcanic tremor (NVT) locations in the Mexican subduction zone: finding the NVT sweet spot. *Geochemistry, Geophysics, Geosystems (G3)*, doi:10.1029/2011GC003916, 2012.
- Maury J., S. Ide, V. M. Cruz-Atienza and V. Kostoglodov. Spatio-temporal variations in slow earthquakes along the Mexican subduction zone. *Journal of Geophysical Research*, doi:10.1002/2017JB014690, 2018.
- Maury, J., S. Ide, V. M. Cruz-Atienza, V. Kostoglodov, G. González-Molina and X. Pérez-Campos. Comparative study of non-volcanic tremor locations: characterization of slow earthquakes in Guerrero, Mexico. *Journal of Geophysical Research*, 121, doi:10.1002/2016JB013027, 2016.
- Villafuerte, C. and V. M. Cruz-Atienza. Insights into the Causal Relationship between Slow Slip and Tectonic Tremor in Guerrero, Mexico. *Journal of Geophysical Research*, 122, doi:10.1002/2017JB014037, 2017.

Figure Rise of tectonic tremor activity in Oaxaca following both the Mw8.2 and Mw7.2 earthquakes. (After Husker et al., GJR, 2019).

Figure Tectonic tremor triggering in Jalisco (northwest of Guerrero) just after the Mw8.2 earthquake (After Miyasawa and Santoyo, 2020)

2. My second comment concerns the general story. Because of the complexity and large number of events, the timeline and the proposed interactions between the elements of the cascade of events are hard to follow. I suggest the authors to add a summary figure which will recap the timing of each event and the proposed interactions (triggering, etc.). This will help the reader to understand the big picture and which interaction (static/dynamic triggering, etc.) have produced with effects along the Mexican subduction zone.

- We perfectly understand this concern, which was also pointed out by another reviewer. To this purpose, largely justified, we have done three things. First, we have coded all aseismic events (e.g. G-SSE1, G-SSE2, O-SSE1,...) across the manuscript and included that terminology in different figures (e.g. Figs. 2, 4 and 6). Then we have created a movie of the sequence where all aseismic events and earthquakes can easily be followed up, and added it as a supplementary material (we strongly suggest the reviewer to watch the movie). Besides, we also included a new figure, Figure 3, where the spatial and temporal evolution of the aseismic slip sequence (and earthquakes) can also be seen at once together with the type of interaction between the events. We have also included the appropriate figure references in the main text and defined the name codes (section "Plate Interface Aseismic Slip History") so that the reader could see both, the movie and Figures 3. We hope this will satisfy the reviewer request.

Other comments:

Page 6 line 13: change 'Fig. 2A' by 'Fig. 2B'.

- Done

Page 7 line 1: add 'help' before 'initiate'

- Done

Page 7 line 6: 'Around the hypocentral region there is a clear rise of CFS...' There is definitively a rise of CFS around the hypocenter of the M7.2 earthquake. However, most of the rise occurs west of Oaxaca in the region that will slip aseismically after the earthquake. I may have missed it, but I don't think this is properly stated in the text. Is there any changes in the seismicity/tremors associated to this rise in CFS?

- It is indeed true that induced CFS in the 2018 Guerrero SSE (G-SSE2) area are high, and therefore it is important to mention. In the section "Plate Interface Dynamic Perturbations" we have thus pointed that out referring Figure 5A. After the study (i.e., after the first submission), we carried out an extensive search for increased seismicity in the SSEs regions with the "match filter" technique. However, due to the low seismicity at these depths (i.e., lack of templates) we have not been able to characterize significant variations in the rate of seismicity so far.

Page 14 line 11: why 0.75 and $L=40\text{km}$. Which comprehensive resolution tests did you performed? In general, it would be great to add a sentence about all the performed tests and where to find the results. This may come from the fact that the methodological is currently under review and therefore not published yet.

- All resolution tests are thoroughly done and analyzed in the accompanying paper (attached to the submission) by Tago et al., (GJI, 2020), which is already under minor revision. We already refer to this work in the manuscript that is also published as a preprint (<https://doi.org/10.1002/essoar.10503378.2>). The two figures below show some results taken from this work that justify the choice of both parameters indicated (Hurst exponent and autocorrelation length of the von Karman function), which determine the regularization of the inverse problem and guarantee the highest slip restitution index across the whole plate interface. Since every line we add goes beyond the allowed space in this journal, we leave the justification already included in the section "Elastostatic adjoint inversion" and refer the reader to the article above.

Figure (after Tago et al., 2020): Synthetic inversion results from the strongly perturbed (noisy) displacements (panel B). The second row of each panel shows the distribution of the restitution index over the plate interface without regularization and for different values of the correlation length, L .

Figure (after Tago et al., 2020). Checkerboard inversions for PS of (B) 80 and (C) 100 km, and correlation length, L , of 20 km. The inverted slip along with the surface displacement fits (left column) and the associated restitution index (right column) are displayed on the 3D plate interface (gray contours). Green triangles are the GPS stations.

Page 15 line 18: The GPS time series are restricted to October 23, 2016 to November 22, 2018. I would be helpful to add longer example time series for a typical station in Oaxaca and Guerrero. This could be added in supplementary material. Such figure will also highlight the unusual analyzed sequence compare to what is typical observed in Mexico.

- This is a very valuable observation for which we are grateful. Consequently, we have created a new figure in the main text (Figure 8) where we present two of the longest displacement time series recorded in Mexico (CAYA and PINO GPS stations) along with others that also allow highlighting the great disturbance of the SSEs cycle in both Guerrero and Oaxaca from the great Mw8.2 Tehuantepec earthquake. The text corresponding to this figure was included in the Discussion of the article.

Page 16 line 6: replace 'as' by 'a' before 'random'.

- Done

Page 17 line 7: why 30km around the Pinotepa Eq and not 50km or 20km?

- Two reasons. (1) Because a radius of 30 km is similar to the epicentral distance of the nearest station (PNIG), which guarantees the greatest number of detections, and (2) because 30 km is less (but comparable) to the expected radius of an earthquake of Mw7.2. A smaller radius would not allow the concentration of seismicity near the hypocenter to be clearly identified (Fig. 5A).

Page 17 line 13: How were the 394 events selected? Is it the whole catalogue? Only a selection? If it's a selection what are the selection criteria?

- These events correspond to previously identified repeating earthquakes detected within the 30km of the epicenter of the Pinotepa earthquake between 2014 and 2019. We chose these events since one of the initial targets of the project was to investigate variations in the repeating earthquake activity and we knew these events were clearly seen as repeater events by at least two stations in the region. As the first technique does not require a strong correlation between the template and the detection, and the templates were free to move around a search area, it was possible to complement and test the results obtained by the two techniques using different sets of templates. We have clarified this in the text.

Page 20 line 6: replace 'stain' by 'strain'.

- Done

Figure 3: It looks like there is some perturbations in the time series of the spontaneous SSE (right on the Figure) associated to the earthquakes (both the M8.2 and M7.2 events). Is there really something there?

- Yes, it is absolutely true. These changes in displacement trends (which are very clear in some cases) reveal how the aseismic slip at the interface, which was already occurring before the earthquakes, accelerated significantly as a consequence of either the radiated waves or the coseismic change of the stresses, depending on which station and which event it is (see the improved Figure 4 and the new Figure 8).

Figure 3A: can you add some GPS stations in Guerrero? I'll be curious to see the evolution of the first SSE in Guerrero even if it occurs mainly before the M8.2 earthquake.

- The time series we present in that figure were carefully selected to illustrate the crustal elastic rebound produced by the Mw8.2 earthquake. GPS time series of the 2017 Guerrero SSE can be seen in Figures 4B, S1 and the new Figure 8.

Figure 3B: On the map there is no GPS stations plotted east of the 7.2 earthquake. If stations are available, it will be interesting to include them to investigate the eastward aseismic slip.

- We understand this point. However, those stations east of the earthquake were already integrated (i.e., investigated) in all inversions that mapped the whole evolution of the post-seismic relaxation and the 2019 Oaxaca SSE, shown in Figures 1 and S2, and now in the new Figure 8, where stations TNNP is also shown. Including them in Figure4B would obscure the main message of the figure.

Figure 5: It's very hard to distinguish the black and green lines. Please change the green color in red or something else.

- Done. We have tried different colors and we are finally left with a lighter green.

Best Regards

REVIEWERS' COMMENTS

Reviewer #1 (Remarks to the Author):

I continue to believe this is a valuable contribution and should be published in NC. The authors have done a good job in addressing my concerns. The story is complicated and they do a good job in presenting it, despite this.

I think it is useful to clarify in discussion that the M8.2 event induced nonlinear elastic conditioning of the crust and plate interfaces, disrupting slow slip and triggering the M7+ events. Alternatively, the associated slow slip changes led to the triggering.

And I'd suggest you define 'conditioning' for geoscientists: Conditioning, a nonlinear elastic memory effect, is a material modulus softening due to dynamic strain perturbation. In the fault core, the granular gouge is preferentially affected by the conditioning in contrast to the fault blocks. This is due to the higher intrinsic mechanical damage of the fault core, and the highly nonlinear nature of granular materials in general. In addition, fluid pressures may drive the effective pressure lower in the fault core, making the core even more susceptible to the conditioning and failure phenomenon.

a few additions/changes:

Line 10—eliminate 'beating'

Fig. 2A-- include M8.2 event

Same, figure 8.

Reference 41 needs title and authors, and earth arXiv ID.

Reviewer #2 (Remarks to the Author):

I'd like to thank the authors for the very thorough, professional and succinctly convincing responses to my original review. I think the paper is much improved for changing the naming system of the SSEs and is much easier to follow. I really like the addition of the timeline in Figure 3, which summarises the sequence of events nicely.

Overall I'm very happy with this revision, and am happy for the paper to be accepted.

Reviewer #3 (Remarks to the Author):

Review comments for NCOMMS-20-33002-R1

This second version of the manuscript by Cruz-Atienza et al. presenting a description of the interactions between SSEs and earthquakes along the Mexican subduction zone during an extraordinary sequence that started with the Mw8.2 Tehuantepec earthquake in September 2017, has been largely improved since the first submission. I want to particularly thank the authors for the great efforts made to improve the readability of the paper with proper labels of the events and a clear timeline presented on Figure 3. I find the story much easier to follow in this new version which makes the whole study more convincing. In addition, I really appreciate the inclusion of long GPS time series on Figure 8. This Figure 8 highlights very clearly the change of behavior on the Mexican subduction interface after the Mw8.2 Tehuantepec earthquake. This was clearly missing in the previous version of

the manuscript. In conclusion, I believe this paper is an excellent work that deserve quick publication. I still have a couple of very minor comments before the paper can be published.

Minor comments:

Page 5 line 7: "Panel A shows that the G-SSE1 basically ended with the occurrence of the devastating Mw8.2 Tehuantepec and Mw7.1 Puebla-Morelos earthquakes". This is also clear visible on Panel B with the ARIG GPS time series. I suggest to add something like: "Panel A (and ARIG GPS time series in Panel B left)..."

Paragraph finishing at Page 13 line 22: I really like this addition to the discussion. Figure 8 really helps to understand the big picture and the extraordinary sequence presented in the paper. Such change in the recurrence behavior of SSEs on a subduction is obviously a complex phenomenon that incorporate both the strong perturbation by the Mw8.2 Tehuantepec earthquake and the frictional properties at the subduction interface as discussed by the authors. I just want to point out that such long-term changes in the creep slip evolution also appears with very simple simulations of the Guerrero subduction zone with a planar interface governed by space-varying static/kinetic friction and dislocation creep (Zigone et al., 2015). Even if this simple model is not realistic, it still shows that variations of friction properties and occurrence of large events on a plate interface can change the evolution of slow aseismic creep over long period of time (~20-30 years). Such variations visible in long-term simulations highlight the need to better understand the history-dependent rheological and frictional properties along with the feedback mechanisms between large earthquakes and SSEs. The work presented by the authors is a very nice step in that direction.

Zigone, D., Y. Ben-Zion & M. Campillo (2015), Modeling non-volcanic tremor, slow slip events and large earthquakes in the Guerrero subduction zone (Mexico) with space-variable frictional weakening and creep, *Geophys. J. Int.*, 202, 653–669, doi : 10.1093/gji/ggv174.

Figure 3 in supplementary caption (page 46 line 2-4): The caption is confusing. In the caption it's written (A-D) 30-day time windows aseismic slip inversions of the G-SSE1 (left column) and the associated cumulative total CFS over the intermediate-depth normal fault where the Mw7.1 Puebla-Morelos earthquake took place on September 19, 2017 (right column)". This suggest that the slip inversion is in the left column and the CFS in the right column. Of course, the authors want to point that each column from left to right shown both slip inversion (top figure) and CFS evolution (bottom figure) for various time windows before the Mw7.1 Puebla-Morelos earthquake. I suggest clarifying the caption to avoid confusion.

Best Regards

Answers to the Reviewers' Comments

- **Reviewer #1 (Remarks to the Author):**

I continue to believe this is a valuable contribution and should be published in NC. The authors have done a good job in addressing my concerns. The story is complicated and they do a good job in presenting it, despite this.

Answer: We thank you for your appreciation of our work and the time you spent reviewing it, which allowed significant improvements in the manuscript.

I think it is useful to clarify in discussion that the M8.2 event induced nonlinear elastic conditioning of the crust and plate interfaces, disrupting slow slip and triggering the M7+ events. Alternatively, the associated slow slip changes led to the triggering.

Answer: We believe this already has been said in pages 13 (20-22) and 14 (lines 14-15).

And I'd suggest you define 'conditioning' for geoscientists: Conditioning, a nonlinear elastic memory effect, is a material modulus softening due to dynamic strain perturbation. In the fault core, the granular gouge is preferentially affected by the conditioning in contrast to the fault blocks. This is due to the higher intrinsic mechanical damage of the fault core, and the highly nonlinear nature of granular materials in general. In addition, fluid pressures may drive the effective pressure lower in the fault core, making the core even more susceptible to the conditioning and failure phenomenon.

Answer: This has been better addressed in the discussion as suggested.

A few additions/changes:

Line 10—eliminate 'beating'

Answer: Done.

Fig. 2A-- include M8.2 event

Answer: As shown in Figure 1, the epicenter of the M8.2 event is far from the area of interest, so including it would mean reducing the scale of the figure, making it difficult to interpret.

Same, figure 8.

Answer: Done.

Reference 41 needs title and authors, and earth arXiv ID.

Answer: Done.

- **Reviewer #2 (Remarks to the Author):**

I'd like to thank the authors for the very thorough, professional and succinctly convincing responses to my original review. I think the paper is much improved for changing the naming system of the SSEs and is much easier to follow. I really like the addition of the timeline in Figure 3, which summarises the sequence of events nicely.

Overall I'm very happy with this revision, and am happy for the paper to be accepted.

Answer: We thank you very much for your appreciation of our work and the time you spent reviewing it, which allowed significant improvements in the manuscript.

- **Reviewer #3 (Remarks to the Author):**

This second version of the manuscript by Cruz-Atienza et al. presenting a description of the interactions between SSEs

and earthquakes along the Mexican subduction zone during an extraordinary sequence that started with the Mw8.2 Tehuantepec earthquake in September 2017, has been largely improved since the first submission. I want to particularly thank the authors for the great efforts made to improve the readability of the paper with proper labels of the events and a clear timeline presented on Figure 3. I find the story much easier to follow in this new version which makes the whole study more convincing. In addition, I really appreciate the inclusion of long GPS time series on Figure 8. This Figure 8 highlights very clearly the change of behavior on the Mexican subduction interface after the Mw8.2 Tehuantepec earthquake. This was clearly missing in the previous version of the manuscript. In conclusion, I believe this paper is an excellent work that deserve quick publication. I still have a couple of very minor comments before the paper can be published.

Answer: We thank you very much for your appreciation of our work and the time you spent reviewing it, which allowed significant improvements in the manuscript.

Minor comments:

Page 5 line 7: “Panel A shows that the G-SSE1 basically ended with the occurrence of the devastating Mw8.2 Tehuantepec and Mw7.1 Puebla-Morelos earthquakes”. This is also clear visible on Panel B with the ARIG GPS time series. I suggest to add something like: “Panel A (and ARIG GPS time series in Panel B left)....”

Answer: Done.

Paragraph finishing at Page 13 line 22: I really like this addition to the discussion. Figure 8 really helps to understand the big picture and the extraordinary sequence presented in the paper. Such change in the recurrence behavior of SSEs on a subduction is obviously a complex phenomenon that incorporate both the strong perturbation by the Mw8.2 Tehuantepec earthquake and the frictional properties at the subduction interface as discussed by the authors. I just want to point out that such long-term changes in the creep slip evolution also appears with very simple simulations of the Guerrero subduction zone with a planar interface governed by space-varying static/kinetic friction and dislocation creep (Zigone et al., 2015). Even if this simple model is not realistic, it still shows that variations of friction properties and occurrence of large events on a plate interface can change the evolution of slow aseismic creep over long period of time (~20-30 years). Such variations visible in long-term simulations highlight the need to better understand the history-dependent rheological and frictional properties along with the feedback mechanisms between large earthquakes and SSEs. The work presented by the authors is a very nice step in that direction.

Answer: We appreciate the suggested reference. Although it is an interesting work, the assumed model does not integrate possible variations in the frictional (constitutive) properties of the fault, potentially produced by the passage of seismic waves. In that sense, our computational simulations seem to be more realistic (but also insufficient) by incorporating a R&S friction law subject to realistic dynamic perturbations. While seismic waves significantly affect the development of an SSE (as shown in the manuscript), we find no evidence of a change in the recurrence period of future events. As you indicate, more sophisticated dynamic models that couple R&S friction with nonlinear elastic effects on the fault core are needed.

Zigone, D., Y. Ben-Zion & M. Campillo (2015), Modeling non-volcanic tremor, slow slip events and large earthquakes in the Guerrero subduction zone (Mexico) with space-variable frictional weakening and creep, *Geophys. J. Int.*, 202, 653–669, doi : 10.1093/gji/ggv174.

Figure 3 in supplementary caption (page 46 line 2-4): The caption is confusing. In the caption it's written (A-D) 30-day time windows aseismic slip inversions of the G-SSE1 (left column) and the associated cumulative total CFS over the intermediate-depth normal fault where the Mw7.1 Puebla-Morelos earthquake took place on September 19, 2017 (right column)". This suggest that the slip inversion is in the left column and the CFS in the right column. Of course, the authors want to point that each column from left to right shown both slip inversion (top figure) and CFS evolution (bottom figure) for various time windows before the Mw7.1 Puebla-Morelos earthquake. I suggest clarifying the caption to avoid confusion.

Answer: Done.

Best Regards